# Spatially Referenced Bayesian State-Space Model of Total Phosphorus in western Lake Erie

Timothy J. Maguire[1], Craig A. Stow[2], Casey M. Godwin[1]

[1]Cooperative Institute for Great Lakes Research, School for Environment and Sustainability, University of Michigan, Ann Arbor, Michigan 48109, United States
[2]NOAA Great Lakes Environmental Research Laboratory, Ann Arbor, Michigan, 48108, United States

*Correspondence to*: Timothy J. Maguire (maguiret@umich.edu)

**Abstract.** Collecting water quality data across large lakes is often done under regulatory mandate, however it is difficult to connect nutrient concentration observations to sources of those nutrients and to quantify this relationship. This difficulty arises from the spatial and temporal separation between observations, the impact of hydrodynamic forces, and the cost involved in discrete samples collected aboard vessels. These challenges are typified in Lake Erie, where binational agreements regulate riverine loads of total phosphorus (TP) to address the impacts from annual harmful algal blooms (HABs). While it is known that the Maumee River supplies 50% of the nutrient load to Lake Erie, the details of how the Maumee River TP load changes Lake Erie TP concentration have not been demonstrated. We developed a hierarchical spatially referenced Bayesian state-space model with an adjacency matrix defined by surface currents. This was applied to a 2km-by-2km grid of nodes, to which observed lake and river TP concentrations were joined. The model generated posterior samples describing the unobserved nodes and observed nodes on unobserved days. We quantified the impact plume of the Maumee River by experimentally changing concentration data and tracking the change of in-lake predictions. Our impact plume represents the spatial and temporal variation of how river concentrations correlate with lake concentrations. We used the impact plume to scale the Maumee River spring TP load to an effective Maumee River TP spring load for each node in the lake. By assigning an effective load to each node the relationship between load and concentration is consistent throughout our sampling locations. A linear model of annual lake node mean TP concentration and effective Maumee River load estimated that, in the absence of the Maumee River load, lake concentrations at the sampled nodes would be 23.1 µg l$^{-1}$ (± 1.75, 95% credible interval, CI) and that for each 100 tons of spring TP effective load delivered to Lake Erie, mean TP concentrations increase by 11 µg l$^{-1}$ (± 1, 95% CI). Our proposed modelling technique allowed us to establish these quantitative connections between Maumee TP load and Lake Erie TP concentrations which otherwise would be masked by the movement of water through space and time.

## 1 Introduction

In a collective response to the economic, human health, and environmental damage caused by pollution, assessing water quality is a regulatory mandate across many waterbodies. In many aquatic ecosystems nutrient concentrations are a primary water quality analyte collected. Observed concentrations are driven by both point and non-point sources. E.g., Wastewater effluent

and excessive nutrient export primarily from agricultural watersheds may lead to eutrophication, harmful algae blooms (HABs), and threatens drinking water contamination (Brooks et al., 2016; Mellios et al., 2020; Schneider and Bláha, 2020). Individual water bodies present data collection challenges, particularly large lakes. Even for those locations with well-funded sample collection schema, trying to describe the spatiotemporal heterogeneity in nutrients is difficult. Discernible trends are

difficult to assess as samples represent discrete spatial data within a system of constantly moving water and asymmetrical spatial extents of riverine nutrient plumes.

Lake Erie is an example of a waterbody that is challenging to model (Ho and Michalak, 2017; Steffen et al., 2017; Stow et al., 2015). While Lake Erie is large (25,700 km$^2$), its western basin is relatively shallow (mean depth 6 m) (Bolsenga and

Herdendorf, 1993) and intense nutrient export from the agriculturally dominated Maumee River watershed leads to episodic HABs (Watson et al., 2016). Commercial fisheries, drinking water, and human health within Lake Erie are all impacted because of the combination of nutrient addition, HABs, and physical lake properties (Wituszynski et al., 2017). Because of these intersecting concerns, a binational effort to regulate phosphorus entering Lake Erie has been active since 1978 (GLWQA, 2012). Nutrient concentration and physical lake data are pivotal in understanding the causes of western basin Lake Erie water

quality issues and have been collected by a broad range of federal, state/province, and local agencies throughout western Lake Erie (Fig. 1). The goal has been to collect these data at a spatial and temporal scale which should lead to a defined relationship of how river nutrient load affects lake nutrient concentration; understanding of the influence of riverine load through time and space; and ultimately the ability to predict how river load reductions would manifest as altered lake concentrations.

Yet, while western Lake Erie is routinely monitored and the nutrient concentration and flow are estimated daily in rivers, a generalizable connection between Maumee River phosphorus load and Lake Erie phosphorus concentrations remains undefined (i.e., if phosphorus load increases by 100 tons what is the response in lake concentration?) (Rowland et al., 2019). Spring Maumee River soluble reactive phosphorus export correlates with western Lake Erie HABs extent; this pattern has been observed since the soluble reactive phosphorus loads started to increase in the 1990s (Ho and Michalak, 2017; Michalak

et al., 2013; Stow et al., 2015). The challenge is that nutrients in the lake move with the water currents, resulting in a complex relationship of upstream and downstream current dependence. Moreover, within-lake phosphorus cycling is dynamic and impacted by biological and physical processes (Li et al., 2021; Matisoff et al., 2016). Additionally, the time between sampling events within this time-series and the size of the lake-river system where models need to be applied inherently adds uncertainty and reduces the predictive efficacy of transport models linked with hydrodynamic models (Schwab et al., 2009).

Bayesian state-space models have been used in ecology to incorporate temporal and spatial autocorrelation and quantify observation error separate from the error attributable to the modelled ecological process (Auger-Méthé et al., 2021; Durbin and Koopman, 2012; Shumway and Stoffer, 2019). State-space models are widely used in ecology to model animal populations (Buckland et al., 2004), movement (Royer et al., 2005), and fisheries stocks (Meyer and Millar, 1999). Bayesian inference can

quantify uncertainty in the effect of nutrient load on nutrient distribution within a dynamic system such as Lake Erie. Non-stationary time-series models have been used in the Great Lakes to model water levels (Lamon and Stow, 2010; Sellinger et al., 2008) and to predict polychlorinated biphenyls concentration in trout (Stow et al., 2004). The goal of this study was to use a spatially dependent time-series state-space model approach to define concentrations at unobserved locations and quantify the impact of river nutrient delivery across western Lake Erie. While spatial models have been used in the Great Lakes for predicting HABs biomass, HABs extent, and nutrient transport (Fang et al., 2019; Schwab et al., 2009), we proposed a Bayesian framework for similar spatial data. We showed that phosphorus concentration, along with an informed uncertainty, can be estimated by state-space models that incorporate concentration data from within the lake, the rivers, and lake surface currents. Although this approach is informed by the currents, it is does not include all the explicit biogeochemical and physical processes that are part of mechanistic models (Rowe et al., 2019). Our contribution to the ecological state space model methodology is the incorporation of a surface current derived adjacency matrix, combined disparate agency field data, and inclusion of data from rivers as sources of information in fitting estimated values within a system dominated by missing values. Together our contribution will fit values in the absence of observations and allow experimentation in archival data previously not possible. Here, we quantified how well our model fits the data and generated predictions of total phosphorus (TP) concentrations across western Lake Erie. TP includes the dissolved and particulate forms of phosphorus. Additionally, we experimentally manipulated observed concentrations to estimate the spatial and temporal impact from the Maumee River plume. Using this delineated Maumee River impact in time and space we tested the hypothesis that when water movement is incorporated, there is a linear relationship between river load and western Lake Erie water TP concentrations. 2 Methods

### 2.1 Study Area and Data Curation

We limited the model spatial window to western Lake Erie (bounded by the portion of the lake west of -83.1° W, Fig. 1) which left ~600 km2 to be defined. We gathered surface concentrations of TP (μg/l) from publicly available databases through Environment Climate Change Canada's Offshore Water Quality Survey, the U.S. Environmental Protection Agency's Great Lakes National Program Office, the Canadian Ministry of the Environment, Conservation and Parks Great Lakes Intake Program, the U.S. National Oceanographic and Atmospheric Administration (NOAA) Great Lakes Environmental Research Laboratory (GLERL) Ecosystem Dynamics Long-Term Research program, and NOAA GLERL Western Lake Erie (WLE) Sampling (Table A1). The data used here extended from 2008 to 2018. For riverine TP concentrations from the Maumee River and River Raisin across the 2008 to 2018 interval we downloaded data from the National Center for Water Quality Research (NCWQR) at Heidelberg University (Table A1). When multiple samples were collected from a node on a single day the sample average was used.

### 2.2 Model Description

We created a model where day $t$ TP concentrations are predicted based on the concentrations "upstream" at day $t-1$. The spatial adjacency of "upstream" relationships was defined by the direction and magnitude of surface currents.

To build our adjacency matrix we first defined a hypothetical distance and direction that surface water moved based on surface currents. We used surface current data retrieved from the NOAA Great Lakes Coastal Forecasting System (GLCFS, Table A1) database. These data are defined by hourly eastward and northward water velocity (m s$^{-1}$) predicted across Lake Erie on a 2 km-by-2 km grid (Fig. 1). Hourly northward and eastward velocity (m day$^{-1}$) for each node for years 2008 to 2018 defined surface current direction in radians (dLat and dLon) using the node latitude (Lat$_0$) and longitude (Lon$_0$), the Earth's radius (R, 6378137 m), the northward velocity offset in meters (dN), and eastward offset in meters (dE) (Eqs 1 and 2). The direction the surface water travelled in radians was used to determine the latitude (Lat$_1$) and longitude (Lon$_1$) which represented by each hourly movement (Eqs 3 and 4), and was repeated for 24-hours until the final position of the surface water movement from each node was determined (Fig B1).

$$dLat = \frac{dN}{R} \tag{1}$$

$$dLon = \frac{dE}{R*Cos\left(\pi*\frac{Lat_0}{180}\right)} \tag{2}$$

$$Lat_{t+1} = Lat_t + dLat_t * \left(\frac{180}{\pi}\right) \tag{3}$$

$$Lon_{t+1} = Lon_t + dLon_t * \left(\frac{180}{\pi}\right) \tag{4}$$

The limited model spatial window of western Lake Erie was represented by 254 nodes (Fig. 1). The Lake Erie surface water TP concentrations were associated with their closest nodes of the same 2 km-by-2 km grid nodes used in the surface current datasets. NCWQR concentration data were collected for the Maumee River (41.5º N, -83.712778º W) and the River Raisin (41.960556º N, -83.531111º W) locations ~30 and ~18 km, respectively, inland from Lake Erie. River concentrations were assigned to the node closest to the river mouth. The assumption that these concentrations represent the conditions at the terminus of the rivers adds uncertainty to our modelling, however the spatial extent of this extra uncertainty should end where the Lake Erie TP concentration data begins to inform the model posterior samples.

**2.2.1 State-Space Models**

We constructed hierarchical, spatially referenced Bayesian state-space models for each year to estimate TP concentrations for each node on each day. The data were logged prior to fitting the model and MCMC samples remained in log space until exponentiated for plotting. The temporal range annually was May 20 to October 2, to coincide with the majority of the WLE sampling. The distance between each daily offset surface current location (Lat$_1$, Lon$_1$) and each 2 km-by-2 km concentration node was measured and the node $n$ with the shortest distance defined the adjacency matrix to associate each node $n$ on day $t$ with the node $k$ on day $t$-$1$. The state-space model consists of two models, an observation model of data (y) and a latent state (x) process model.

$$y_{n,t,y} \sim N\left(x_{n,t,y}, \sigma^2\right) \tag{5}$$

Log-transformed TP concentration observations ($y$) at the $n^{th}$ node on the $t^{th}$ day of the $y^{th}$ year was estimated with a normal data model sampled from the unobserved latent state variable ($x$) at the $n^{th}$ node on the $t^{th}$ day of the $y^{th}$ year with standard deviation σ (Eq 5). The process model is a first order Markov, only depending on the value of the node at time *t-1* which transported TP to node $n$ at time $t$, that source node is denoted $k$. For nodes in the river, $k = n$ and for nodes in the lake, $k$, is determined from the time $t$ adjacency matrix.

$$x_{n,t,y} \sim Truncated\ N\left(f\left(x_{n,t-1,y}\right), \tau^2\right)\quad I(a \le x_{n,t,y} \le b) \tag{6}$$

where

$$f\left(x_{n,t,y}\right) = \begin{cases} x_{k,t-1,y} * \beta_{mau} & if\ n = Maumee\ River\ Node \\ x_{k,t-1,y} * \beta_{rai} & if\ n = River\ Raisin\ Node \\ x_{k,t-1,y} * \beta_{self} & if\ n = same\ lake\ Node \\ x_{k,t-1,y} * \beta_{lake} & if\ n = different\ lake\ Node \end{cases} \tag{7}$$

The latent state ($x_{n,t,y}$; Eq 6) is sampled from a normal distribution of a predicted latent state ($f(x_{n,t-1,y})$, Eq 7) and standard deviation τ. $x_{n,t,y}$ was truncated by the detection limit of TP laboratory analysis (5 µg l$^{-1}$, $a$, Eq 6) and the maximum value observed in each year ($y$) within the Maumee River ($b$, Eq 6). The maximum observed Maumee River concentration was used as the upper truncating value because no observation in the lake will exceed this value. Priors were uninformative and defined as;

$\sigma, \tau \overset{iid}{\sim} Gamma(0.001, 0.001)$

$\beta_{self}, \beta_{lake} \overset{iid}{\sim} Normal(0, 10{,}000)$

$\beta` \sim Normal(0, 10{,}000)$

$\tau`_{mau}, \tau`_{rai} \overset{iid}{\sim} Gamma(0.001, 0.001)$

$\beta_{mau} \sim Normal(\beta`, \tau`^2_{mau})$

$\beta_{rai} \sim Normal(\beta`, \tau`^2_{rai})$

River model coefficients ($\beta_{mau}$ and $\beta_{rai}$) were fit hierarchically ($N(\beta`, \tau`^2)$) because the ecological and anthropogenic processes enacted on these watersheds are similar, if at different scales. The two lake models were fit with two independent β coefficients depending on if the nearest adjacent node $k$ is the same as the estimated node $n$ ($\beta_{self}$) or if a different node $k$ is the nearest

($\beta_{lake}$). Separate independent in-lake models were used to capture different potential drivers of TP concentration through time depending on whether each node was subject to little surface water movement ($\beta_{self}$) or active surface water movement ($\beta_{lake}$). In 2012 there were no River Raisin observations and so the model in 2012 treats the River Raisin node as a lake node. The

model was run in R (version 4.0.2) and JAGS (version 4.3.0) (Eddelbuettel, 2017; Microsoft Corporation and Weston, 2020; Plummer, 2019). Each year's model iteration count was 50,000 with a thin of 10, representing 5,000 effective samples along three independent Markov chains. The chain convergence was monitored by Gelman and Rubin's convergence diagnostic. Scale factors less than 1.1 were used to define when chains had converged (Plummer, 2019). Initial conditions for the latent state $x_{n,t=1,y}$ were defined as the mean and variance of the previous year first 20 days. The first year (year = 2008) initial conditions were estimated as N(12, 5) (Rockwell et al., 2005).. The goodness of fit of the models was described via posterior predictive p-values (Gelman, 2013) and Bayesian $R^2$ (Gelman et al., 2019), while the performance between years and across nodes was assessed with K-Fold cross validation (CV) utility (Geisser and Eddy, 1979; Piironen and Vehtari, 2017) (Eq 8).

### 2.2.2 Fitting the SSM and Diagnostics

Posterior predictive p-values were calculated by model year with test statistic mean TP concentration, to compare means of observations to the means of the model outputs. For each year, a posterior p-value distribution was described by 15,000 bootstraps of 100 resamples from the observed node posteriors. Bayesian $R^2$ defined as the fitted variance ($var_{fit}$) divided by the sum of $var_{fit}$ and the residual variance ($var_{res}$) was calculated for each model year. Model $var_{fit}$ was the variance of the modelled predictive mean, while $var_{res}$ is estimated by squared standard deviation of the errors (Gelman et al., 2019).

K-Fold CV utility compared model predictive performance across years and across nodes. The cross-validation via leave one-node-out was used to evaluate model predictions where observations are not available. This estimate of model performance is needed as our dataset has far fewer observations than the product of nodes and timepoints and estimates of $R^2$ report how well the model does while given all the available data. Additionally, the cross-validation estimates aggregated by unobserved node (space) or by year (time), defines how well the model estimates TP values irrespective of the node's proximity to the TP river sources or the number and location of annual in-lake observations. The cross validated utility was applied by removing all $d$ observations from a randomly selected node $k$ with at least 10 observations collected during the model year. K-Fold CV was calculated 3 times per year, for years that had less than 3 nodes with at least 10 observations, all nodes that satisfied the 10-observation cut-off were used. K-Fold CV is the mean leave-one-node-out log predictive density from posterior samples of the omitted $d$ observation $\hat{y}_{k,t,y}$ at node $k$, day $t$, year $y$, were compared to observed concentrations at $y_{k,t,y}$ (Piironen and Vehtari, 2017) (Eq 8).

$$K - \text{Fold CV}_{n,y} = \frac{1}{d}\sum_{d=1}^{d} \log p\left(y_{k,t,y}|\hat{y}_{k,t,y}\right) \tag{8}$$

Using a model that describes posterior predictive distributions of mean K-Fold CV across years and nodes we examined if our state-space approach preferentially generated predictions for certain years or certain nodes that contain the observed values (Eq 9 and 10). 95% credible differences between group means (for nodes $\mu_n$ or for years $\mu_y$) that do not contain 0 were used

to determine if groups were different (e.g., if the 95% credible difference from $\mu_{2018}$ - $\mu_{2017}$ contains 0 these means are not considered different).

$$\text{K} - \text{Fold CV}_n \sim N\left(\mu_{on} + \sum_n^1 \mu_n, \sigma_n^2\right) \tag{9}$$

$$\text{K} - \text{Fold CV}_y \sim N\left(\mu_{oy} + \sum_y^1 \mu_y, \sigma_y^2\right) \tag{10}$$

$\mu_{on}$ and $\mu_{oy}$ were fit with normal priors $\left(N\left(\overline{K - Fold\ CV}, 5 * \sigma_{KFCV}^2\right)\right)$, $\sigma_{KFCV}$ was defined as the standard deviation of the K-Fold CVs. The $\mu_n$ and $\mu_y$'s were given normal priors $\left(N\left(0.1,\ \sigma_\mu^2\right)\right)$, and $\sigma_\mu$ which functions as the within-group variance has a gamma prior with rate and shape estimated from the mode and standard deviation of the K-Fold CVs (Kruschke, 2015 p560). Finally, $\sigma_n$ and $\sigma_y$ which represent the between-group variance were fit with a uniform prior $\left(uniform(100^{-1} * \sigma_{KFCV}, 10 * \sigma_{KFCV})\right)$. $\sum \mu_n$ and $\sum \mu_y$ were constrained to 0 when fitting $\mu_{on}$ and $\mu_{oy}$.

### 2.3 Model Experimentation

Our state-space models were used to test the hypothesis that western Lake Erie TP concentrations are a linear function of Maumee River TP load when surface water movement is incorporated. We incorporate water movement into our linear model by first estimating the spatial impact of the Maumee River. The Maumee River impact plume was estimated by artificially reducing the Maumee River TP concentrations by 50% $\left(\grave{y}_{Maumee,t,y}\right)$, each year's model was then refit (Eq 5-7). The model output for each node was examined and the position of each node's concentration $\left(\grave{y}_{n,t,y}\right)$ 95% predictive intervals (PI) was compared to the original model ($y_{n,t,y}$). The change from the original 95% PI of $y_{n,t,y}$, which we call the deflection, was interpreted as evidence that the Maumee River node was, at some time-step, influencing node $n$. The annual mean root squared sum of the $\grave{y}_{n,t,y}$ 95% PI change compared to the $y_{n,t,y}$ 95% PI was then normalized by the largest value for that model year ($y$), this normalized estimate of PI change ($d_{n,y}$) across the 254 nodes within our spatial window was used to define effective Maumee River spring TP impact within Lake Erie. We estimated Maumee River spring load estimates ($l_y$, tons TP) by multiplying NCWQR daily flow and TP concentration data (Table A1) from March 1 to July 31 annually. Finally, we multiplied $d_{n,y}$ and $l_y$ to represent a spatially explicit effective Maumee River TP spring load at each node $\left(\grave{l}_{n,y}\right)$.

A linear model of mean TP concentration ($\bar{y}_{n,y}$) per year per node ($n$, where node $n$ had at least one observation) as a function of effective spring Maumee River TP load $\left(\grave{l}_{n,y}\right)$ was used to test for a linear relationship between Maumee River load and Lake Erie surface water TP concentrations. The model was fit in a Bayesian framework which allowed us to fit the heteroskedastic relationship of concentration and effective load by fitting a positive linear relationship to model variance and effective load (Eq 11). $\beta_{1,2}$ were given non-informative normal priors ($N(0, \frac{1}{0.001^2})$) while $\alpha_{1,2}$ were given non-informative log normal priors ($logN(0, \frac{1}{0.001^2})$) because they must be positive random variables.

$$\bar{y}_{n,y} \sim N\left(\beta_1 + \beta_2 * \grave{l}_{n,y}, \left(\alpha_1 + \alpha_2 * \grave{l}_{n,y}\right)^2\right) \qquad (11)$$

## 3 Results

The annual data sets defined by TP concentration observations and riverine TP data on our 2-km by 2-km grid in western Lake Erie contained an average of 99.1 % missing values. The number of nodes that contained observations ranged from 14 to 40 among years. The mean number of samples available at each observed lake node during the model year ranged from 2 to 9. Within the 252 Lake Erie nodes across the available 11 years, a total of 1,218 observations were collected; our hierarchical spatially referenced Bayesian state-space model was then able to provide estimates for the 375,774 unobserved TP concentrations. Between the Maumee River and River Raisin, a total of 2,258 observations were available in the dataset and the missing 734 values were also described by posterior distributions. The mean values of the observed TP concentrations within the Maumee River, River Raisin, and western Lake Erie were 170 µg l-1 (95% interval, 3.5 to 438 µg l-1), 80 µg l-1 (95%, 40 to 215 µg l-1), and 38 µg l-1 (95%, 10 to 203 µg l-1), respectively.

### 3.1 State-Space Model Fit

To assess the goodness-of-fit of the model we determined annual Bayesian $R^2$, posterior predictive p-values, and k-fold CV utility. The 11 years of models had mean Bayesian $R^2$ values from 0.84 to ~1 and mean posterior predictive p-values from 0.42 to 0.59 (Table 1). Posterior predictive p-values of 0.5 indicate a good fit between model output and observations and the 95% CI of all our yearly posterior p-value distributions contain 0.5. Finally, the results of the k-fold cross validation utility 95% credible difference showed no difference across all pairwise comparisons of mean K-fold CV by year or node.

### 3.2 State-Space Model Outputs

Posterior distributions for each node on each day provide estimates for TP concentrations where observations are present and in the absence of observations (2018 in Fig. 2, 2008-2018 in Appendix D). Mean and 95% PI model posterior samples of each node at every day defined our predicted concentration. By example, the Maumee River node in 2018 shows the model following the data and widening PIs where observations are missing (Fig. 2a). For Lake Erie nodes that contained observations the posterior samples follow the broad trend in the observed data (Fig. 2b). Nodes without any observations also follow the trend in downstream observed nodes, and while the uncertainty is larger at unobserved nodes the PIs stay within expected values (Fig. 4c).

### 3.3 Model Experimentation

After artificially reducing the Maumee River concentrations by 50%, the nodes where TP concentration PIs were altered were defined as being within the Maumee River area of impact. The mean square root of each node's summed squared deflection

annually normalized by the largest mean value ($d_{n,y}$) in general was highest near the mouth of the Maumee. The impacted area spread south and east along the State of Ohio coast most years, but some years were subject to larger plumes distributed further north (Fig. 3, Video Supplement 1).

The normalized annual mean Maumee impact estimates ($d_{n,y}$) generated per node were used to adjust spring load to an effective spring load $\left(\grave{l}_{n,y}\right)$ at each node where samples were collected. Lake Erie TP concentration was linearly correlated to the effective Maumee River TP spring load (Mean node concentration = 23.1 ($\pm$ 1.75, 95% CI) + 0.11 ($\pm$ 0.01,95% CI) * Effective Spring Load (tons TP); Fig. 4). The heteroskedastic error in the mean concentration ($\bar{y}_{n,y}$) and effective load $\left(\grave{l}_{n,y}\right)$ relationship

was defined by a linear function $\left(\alpha_1 + \alpha_2 * \grave{l}_{n,y}\right)$. $\alpha_1$ was estimated to be 2.9 ($\pm$ 1.4,95% CI) and $\alpha_2$ was 0.04 ($\pm$ 0.008,95% CI).

## 4 Discussion

### 4.1 State-Space Model Fit

By combining the western Lake Erie TP observations with riverine data and surface currents within a Bayesian model
framework we were able to generate estimates of TP across time and space. The models consistently generated plausible posterior samples for mean TP concentration as each 95% CI of annual posterior predictive p-values included 0.5 and annual Bayesian $R^2$ 95% CI values ranged from 0.84 to 0.99 (Table 1). Annual posterior predictive p-values indicate that our model framework is preforming well predicting water quality within large water bodies even with sparse observations within the data. While our high Bayesian $R^2$ values appear to support the use of our model, it is likely that they represent an inappropriate
model metric. The state-space framework forces the model to pass through the observed data and the daily observation in the river datasets (Figure 2a and 2d) are likely driving Bayesian $R^2$ values higher with their constrained predictive intervals. Because of these elevated Bayesian $R^2$ estimates, the k-fold CV are important checks on the applicability of this state-space approach. The k-fold CV results generated by removing all the observations of a randomly selected lake node with at least 10 observations showed that model predictions were equally accurate across years and by node. Predicting equally well across
the nodes and within any year provides strong support for this framework as being a useful application of Bayesian methods in water quality modelling.

TP is a conservative water quality constituent. TP observations are insensitive to biogeochemical transformations of phosphorus form because these data represent both the organic and inorganic forms of phosphorus occurring in the water
column. βmau, βras, βlake, and βself fit in our models had 95% predictive intervals encompassing a value of 1. Coefficients were close to 1 because on our daily time-step, the TP concentrations do not widely vary (e.g., the concentration today is similar to the concentration yesterday). The uncertainty in the process and data models allows the model predictions to trend

toward the observations where available and be constrained where previous time-steps passed through observations. Were these coefficients to exceed 1 this would be evidence of other inputs of P or less that 1 would indicate some internal loss such as settling or dilution. TP is conservative to processes within the water column because it accounts for the dissolved and particulate P, if our model was applied only to dissolved P which is subject to strong assimilation pressure by phytoplankton the model coefficients would likely be negative. While dilution, settling, and internal loading of TP are happening within our modelled extent in western Lake Erie. Our model lacks the specificity to capture dilution, settling, and internal loading and therefore their effect is being accumulated in our error terms. However, this state-space framework could be defined with a mechanistic process model that did capture these effects. Additionally, while our framework could be implemented with the coefficients (βMau, βRas, βLake, and βSelf) fit hierarchically by year potentially defining the overall effect of dilution, settling, and internal loading, current restrictions on computer memory prevented that use here. However, for smaller spatial and temporal models it could be effective.

No identifiability issues were found, this was assessed by visually determining if priors dominated the fit of coefficients (Auger-Méthé et al., 2021) (Figure C1(a) and C1(b)). The lack of σ and τ correlation was also visually assessed (Figure C1(c)). Fit process model or data model uncertainty were well identified (Figure C1(d)). The apportionment of uncertainty between the process model and the data model varied from year to year (Table C1). This was driven by annual variation in the data model uncertainty. The 2012 Raisin River coefficient ($\beta_{ras}$) predictive interval was larger than other years because of a lack of data in that year. The proportion of uncertainty between process and data model also varied only slightly (Table C1), possibly because of the number of or spatial position of observations. We propose these annual differences were due to the combination of the number of samples collected and their relative position to the surface currents. However, the uncertainty within our models did not prevent accurate outputs estimating TP concentrations at observed and unobserved nodes.

### 4.2 State-Space Model Output

An important property of this modelling approach is that the surface current derived adjacency matrix we used to define our predicted spatially explicit latent state concentration ($x_{n,t,y}$) also produced estimates of TP concentration at nodes where no observations were available. This approach takes discrete measurements in western Lake Erie and establishes connections across the lake surface and through the model year. The model does this across 136 days and 254 2km by 2km nodes, yet model uncertainties are within the range of TP concentrations expected for western Lake Erie. Nodes along the eastern perimeter of the spatial window of our model have additional uncertainty inherent in their position. Occasionally, they will not be associated with the proper "down gradient" node because the extent removes those nodes. Within our system there is little practical effect as these nodes are far from the Maumee River and are dominated by low concentrations. This is a potential problem in other systems and may necessitate wider spatial windows to eliminate.

Our model framework allows information from discrete grab samples to be shared across any waterbody where the movement pattern of water is available. Additionally, this model can generate estimates at unobserved nodes or at unobserved time-steps of observed nodes without requiring defined biogeochemical processes of a mechanistic model. For our application, the 2km x 2km grid was chosen to match the surface current dataset. While we did not experiment with other discrete grid distances, any applicable configuration will work. Defining a reasonable grid distance could be based on the spatial distribution of the

available data and the user's willingness to extrapolate or average surface current direction and magnitude. Similarly, our temporal time-step was daily, but this could also be applied to monthly data in data sparse systems or hourly data in data rich applications. This spatial and temporal flexibility or using state space frameworks gives users the capacity to tune the computational runtime and resolution of models to fit the hypothesis tested.

Within Lake Erie, having estimates for unobserved nodes and nodes that are infrequently sampled allows a connection between discrete point data collected by boat and data layers which cover large sections of lake surface. The spatial distance and temporal disconnect between the data generated by multiple actors on the same waterbody often precludes the combined use of data from multiple projects. However, state-space models which explicitly incorporate time as well as space via water movement could harness more of the available data to make predictions beyond the spatial bounds of the original projects. For

example, remote sensed data layers would be especially useful in this model structure, both as predictor variables and as the response variables. Connecting estimates of TP concentrations to chlorophyll-a concentrations would enhance existing predictive models of cyanobacteria distribution and biomass (Fang et al., 2019).

The agencies and organizations collecting grab samples within Lake Erie would also be able to use the state-space model

output to select sample locations specific to hypotheses. The model can be used to predict the movement of high nutrient water masses which investigators could target. Additionally, projects examining the impact of the Maumee River could sample in and out of the Maumee River impact plume. Beyond the impact to field work, this modelling approach can also be used in model selection. Since this modelling approach is based entirely on observations in the absence of independent explanatory variables, it should be used as a benchmark model for future mechanistic or more complex models to be tested against.

**4.3 Model Experimentation**

Having demonstrated the functional capacity of our hierarchical spatially referenced Bayesian state-space model predicting TP concentrations, our hypothesis was that a linear relationship exists between spring Maumee River load and observed Lake Erie concentrations. By experimentally reducing the concentrations for the Maumee River and rerunning the model we were able to track the "downstream" repercussions to the lake node predicted values and infer the Maumee River impact plume.

Tracking a plume of TP impact using the grab samples was not previously possible because of the distances between sampling locations and the fact that the number of unobserved days outnumber the observed days. Distance and direction across our model extent is wrapped up in the change observed in predictive intervals through time. In our framework distance and whether

the water mass from the Maumee River physically moves toward a node combine. There are several days in which even the nodes closest to the Maumee River are bypassed because the currents take Maumee River water in a different direction. The dual complications of distance and movement have complicated previous attempts at defining a single relationship between Maumee River load and observed in-lake concentrations, which we overcome here.

The plume extent in general follows the southern coast (Fig. 3), which would be expected because of the movement associated with the Coriolis effect. Importantly, this is not a plume that displays high concentration of TP, rather this is the impact plume of the Maumee River. Concentrations outside the impact plume are not influenced, or weakly influenced, by the Maumee and thus load reductions within the Maumee River would not impact lake concentrations in those areas. Our linear model (Eq 11) estimated that when the effective load of the Maumee was 0, the mean annual concentration in the area where samples were collected would be 23.1 $\mu$g l$^{-1}$ ± 1.75, 95% CI.

Each year the Maumee River TP impact plume dimension and intensity changed. Rowland et al. (2019) demonstrated how a linear model of Lake Erie TP observations as a function of Maumee River spring loads defined positive relationships at the closest nodes. Here, we were able to fit parameters that define the load to concentration relationship across all western Lake Erie. Much of the regulatory attention in addressing Lake Erie HABs has focused on Maumee River spring export and providing this quantitative connection is important in furthering watershed TP reduction efforts. Our model estimated that for each 100 tons of spring TP effective load delivered to Lake Erie, TP concentrations in the lake increase by 11 $\mu$g l$^{-1}$ (± 1, 95% CI). We could use our defined linear relationship for hindcasting expected concentration reductions in western Lake Erie based on Maumee spring TP loads which were reduced by 40% for all our model years. Additionally, given a mean concentration maximum, we could predict the load reductions required in previous years to meet that target. Using our linear relationship between lake concentration and spring load to make forecasts for future years is harder. The size and shape of the Maumee River impact ($d_{n,y}$) changes each year (Video supplement 1) and our method defines the river impact from observations. Without being provided an estimate of $d_{n,y}$, a forecast of mean western Lake Erie TP concentrations based on a proposed spring TP load is not achievable. An achievable next step for this modelling framework could be linking the size of the Maumee River plume and Lake Erie TP concentrations to HABs biomass and toxin production, the spatial aspect of such a model could explain why the relationship between bloom biomass and Maumee TP export is not linear (Obenour et al., 2014).

## 5 Conclusions

Our state-space model framework was shown to fit the data well, generated reasonable estimates of concentration at observed and unobserved locations, was modified experimentally to estimate a river impact plume, and used the experimentally derived plume to test a regulatory relevant hypothesis. Adequately characterizing water quality in a large waterbody is difficult.

Sampling and laboratory analysis is expensive and excessively time consuming to feasibly cover even a portion of Lake Erie with high temporal and spatial resolution. However, we demonstrate that a Bayesian state-space framework informed by an adjacency matrix defined by surface currents can generate daily TP concentration which are constrained by uncertainties appropriate for lake conditions. By combining the data from two rivers entering the lake our model (Eq 5-7) enables the rivers to inform the observed and unobserved lake nodes, the observed lake nodes inform the unobserved lake nodes, and unobserved

lake nodes also inform unobserved and observed nodes. This information sharing across time and space empowers this model to connect sparse data across large distances. By experimenting with the model, we were able to estimate a plume of impact from the Maumee River and apply the experimental results to hypothesis testing. The model is amenable to using remote sensing data and can effectively connect lake wide datasets with discrete grab samples. The application here used TP, but any analyte could be modelled in this same structure to generate estimates through time and space, hypothesis test, or to build

baseline models to test process-based models against.

## Code and Data availability

Template code for reproducing our model is available publicly on Zenodo; at https://doi.org/10.5281/zenodo.5570508. All the data used here were from publicly available sources which we provide in Appendix A. On the Zenodo site we made our curated data for 2018 available.

## Video supplement

A supplemental video of Maumee River impact plume through time is available through Copernicus.

## Author contribution

TJM, CAS, and CMG designed the models, the model fitting quality control and the model experiments. TJM developed the model code and performed the simulations. TJM prepared the manuscript with contributions from CAS, and CMG.

## Competing interests

The authors declare that they have no conflict of interest.

## Acknowledgements

The authors would like to acknowledge Peter Alsip of University of Michigan Cooperative Institute for Great Lakes Research (CIGLR) for collating the data from sources listed in Appendix A. The authors would also like to acknowledge the field personnel of CIGLR and National Oceanic and Atmospheric Administration (NOAA) Great Lakes Environmental Research Laboratory (GLERL) Western Lake Erie sampling captains and crews whose field work inspired this manuscript. This publication is contribution #1188 from CIGLR and contribution #1992 from NOAA GLERL. Funding was awarded to CIGLR through the NOAA Cooperative Agreement with the University of Michigan (NA17OAR4320152). Funding was provided by the Great Lakes Restoration Initiative.

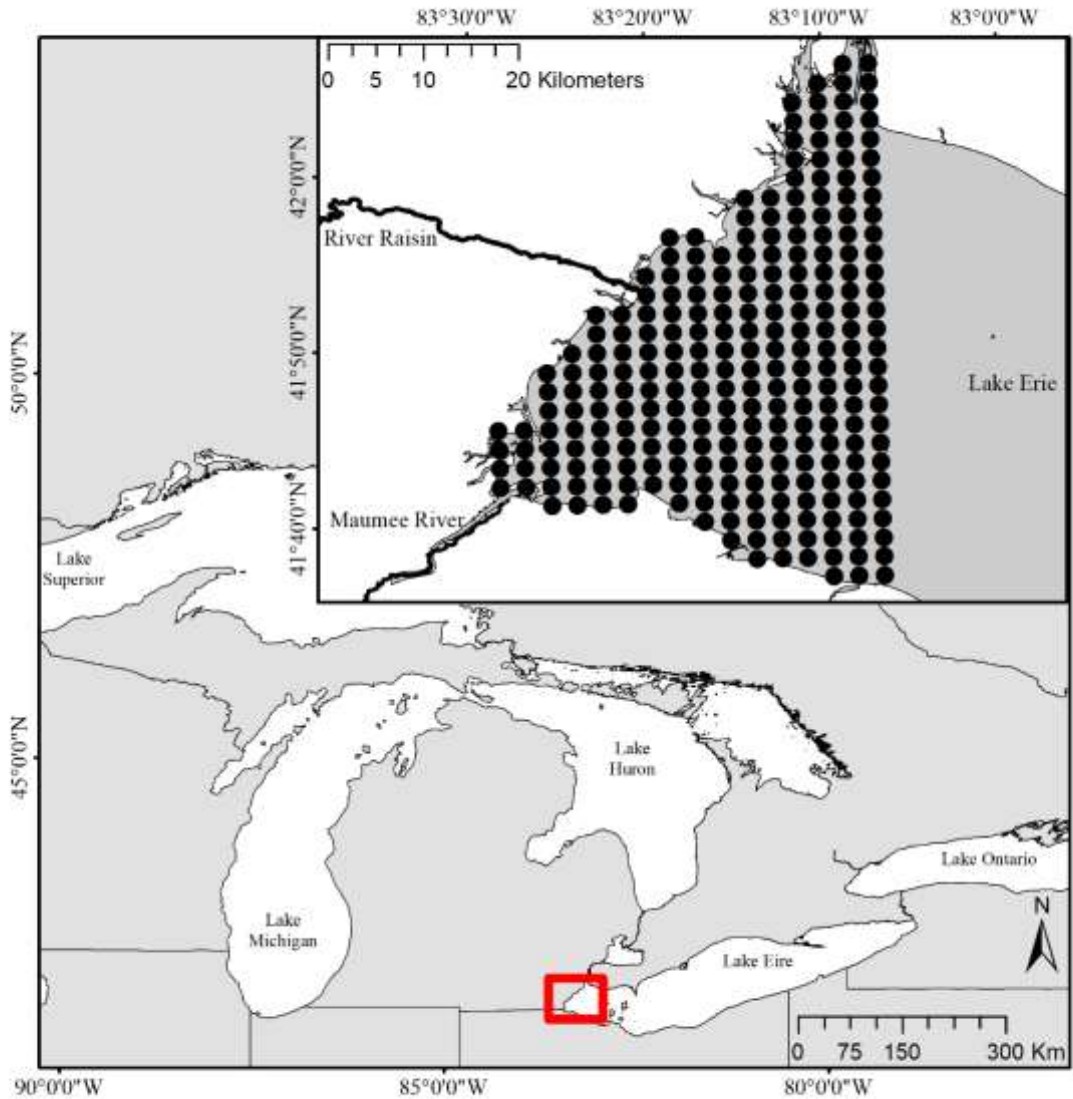

**Figure 1. Map showing the location of study region in western Lake Erie. The inset map shows the tributaries and loke nodes that were included in the model. Our site boundary was defined by the western portions of Lake Erie. A grid of 2 km-by-2 km nodes was used to snap existing concentration data and define an adjacency matrix based on surface currents.**

**Table 1. Bayesian model assessment via p-value (posterior predictive p-values of 0.5 are indicative of a good fit and 95% credible intervals (CI) of our yearly results each containing 0.5) and $R^2$ (each year > 0.8) showed the model generated posterior samples similar in structure to the observations.**

| Year | Posterior Predictive p-values | | $R^2$ | |
|------|------|------|------|------|
| | 95% CI | | 95% CI | |
| 2008 | 0.4 | 0.6 | 0.987 | 0.999 |
| 2009 | 0.49 | 0.68 | 0.965 | 0.978 |
| 2010 | 0.4 | 0.59 | 0.838 | 0.882 |
| 2011 | 0.37 | 0.56 | 0.995 | 1 |
| 2012 | 0.4 | 0.6 | 0.994 | 1 |
| 2013 | 0.32 | 0.51 | 0.914 | 0.974 |
| 2014 | 0.4 | 0.59 | 0.936 | 0.974 |
| 2015 | 0.37 | 0.57 | 0.931 | 0.969 |
| 2016 | 0.41 | 0.6 | 0.925 | 0.962 |
| 2017 | 0.41 | 0.61 | 0.993 | 1 |
| 2018 | 0.39 | 0.59 | 0.995 | 1 |

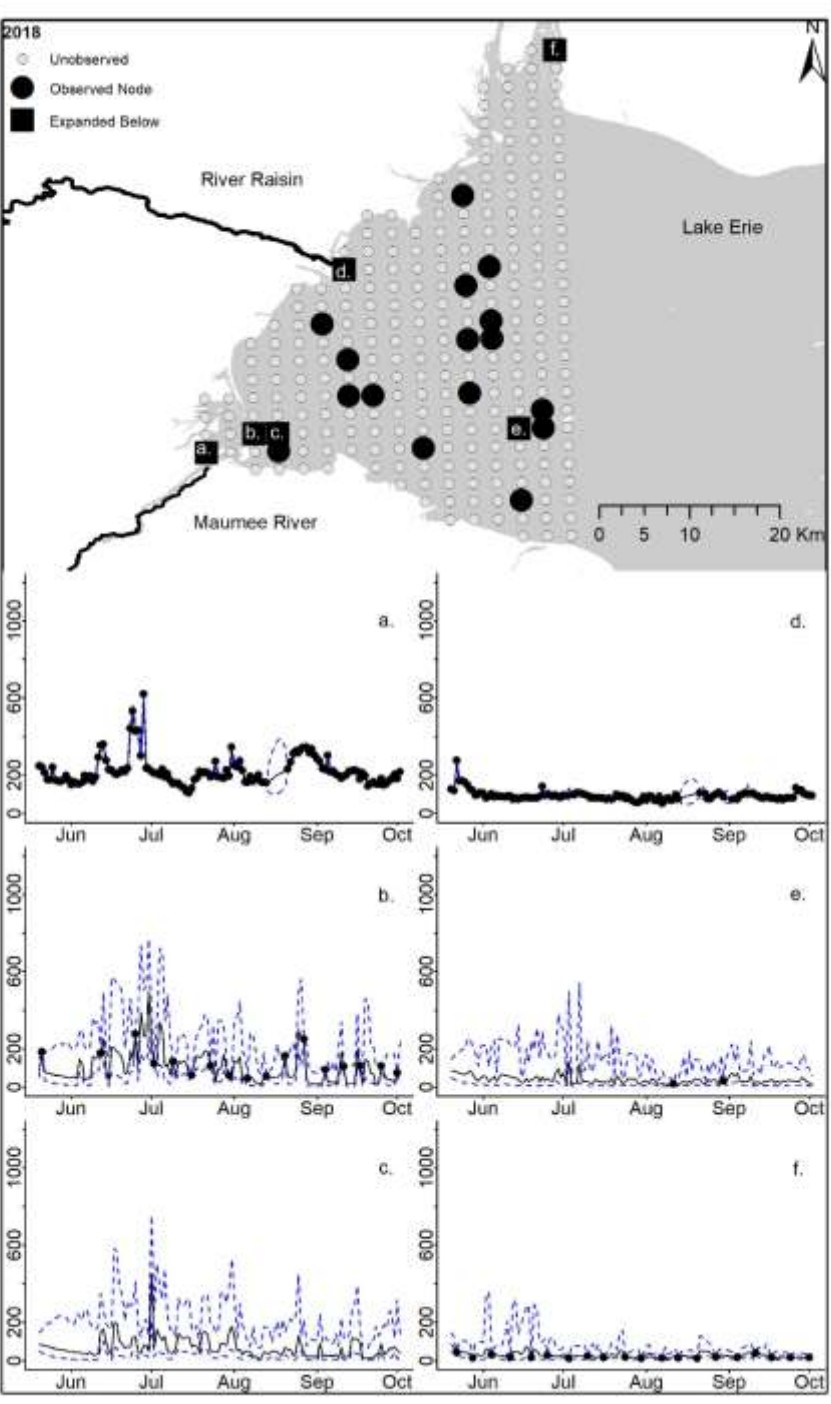

Figure 2. For 2018 the total phosphorus concentration (µg l$^{-1}$) at observed and unobserved nodes were estimated from the model posterior samples. Mean (solid black line) and 95% PI (dashed blue line) for the model posterior samples of each node at every day for (a) the Maumee River, (b,c,e,f) western Lake Erie nodes, and (d) the River Raisin.

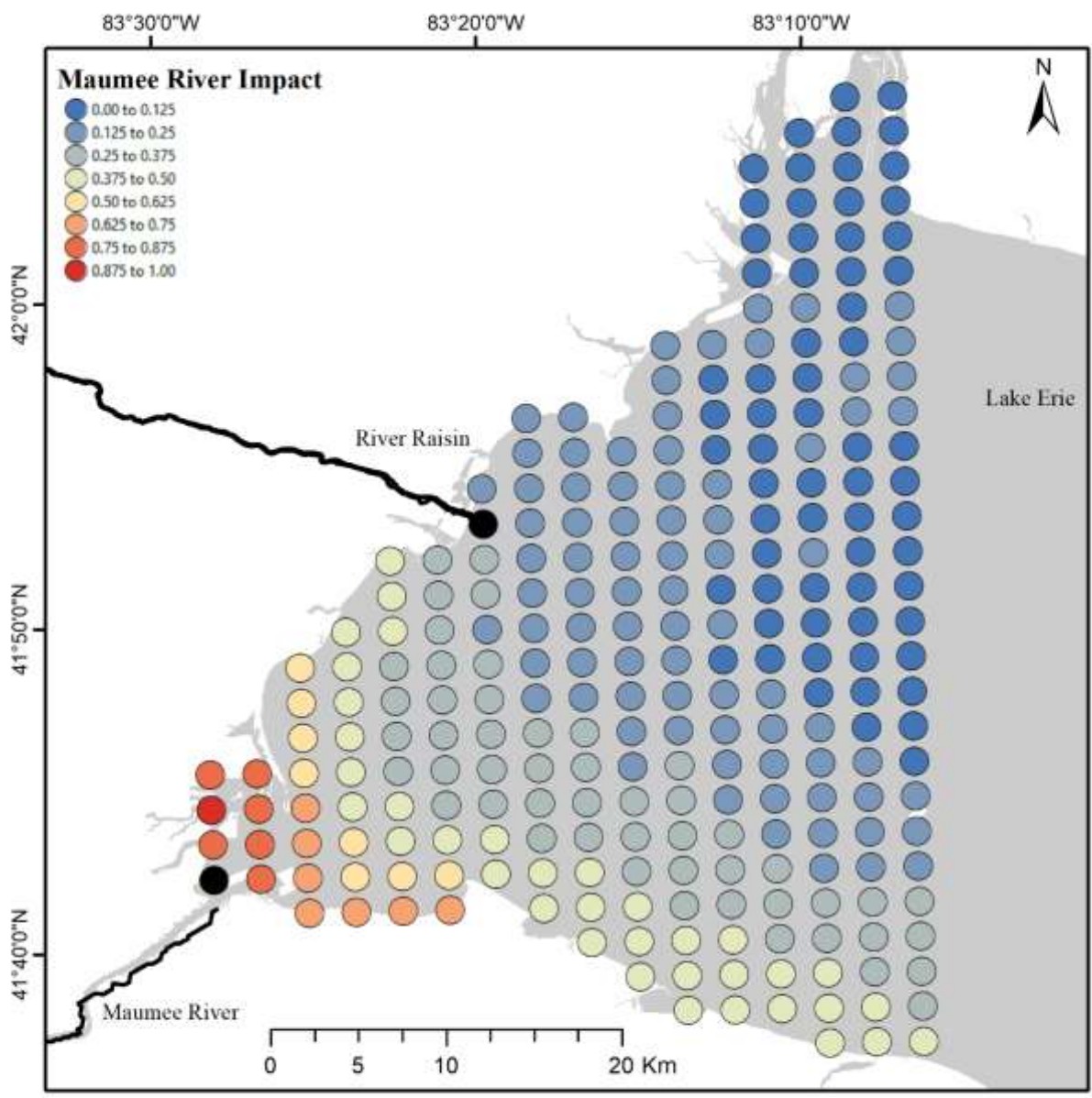

**Figure 3. Heatmap of the mean Maumee River impact plume from 2008 to 2018.**

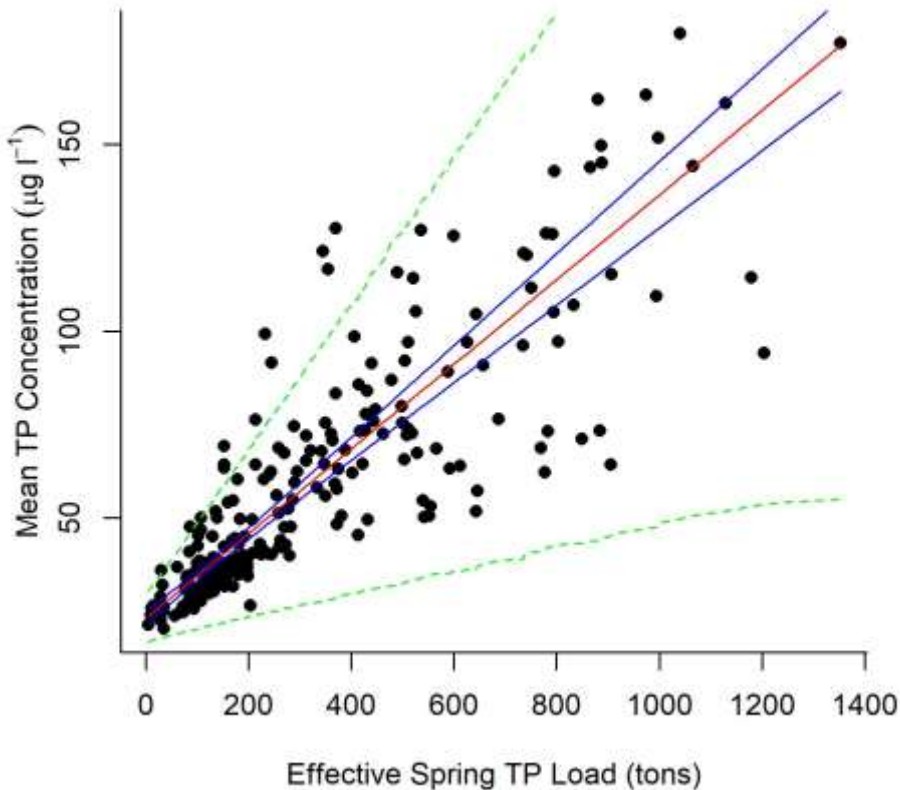

**Figure 4. Mean concentration at the observed nodes for each year was modelled as a function of the relative Maumee River spring TP load (Mean concentration = 23.1 (± 1.75, 95% CI) + 0.11 (± 0.01,95% CI) * Effective Load), where variance in concentration increased linearly with effective load. Effective load was defined by multiplying the normalized river impact generated by experimentally tracing the Maumee River's impact on Lake Erie nodes annually. 95% predictive intervals of the data (green dotted**

**lines) and 95% credible intervals of the linear relationship (blue solid lines) were generated from the model output.**

**Appendices**

**Appendix A**

**Table A1. The data sources for total phosphorus concentrations and surface currents were all retrieved from publicly available online repositories.**

| Agency | Link | Data Type | n (2008 to 2018) |
|---|---|---|---|
| Environment Climate Change | Digital Object Identifier: 10.18164/495eb10d-d423-432a-980f-264ef287d45b | Total Phosphorus Concentration ($\mu g\ l^{-1}$) | 121 |

| | | | |
|---|---|---|---|
| Canada's Offshore Water Quality Survey | | | |
| U.S. Environmental Protection Agency's Great Lakes National Program Office | https://cdx.epa.gov/ | Total Phosphorus Concentration ($\mu$g l$^{-1}$) | 149 |
| Ministry of the Environment, Conservation and Parks Great Lakes Intake Program | http://files.ontario.ca/moe_mapping/ downloads/2Water/GLIP/All_Lakes_GLIP.csv | Total Phosphorus Concentration ($\mu$g l$^{-1}$) | 637 |
| National Oceanographic and Atmospheric Administration (NOAA) Great Lakes Environmental Research Laboratory (GLERL) Ecosystem Dynamics Long-Term Research program | Digital Object Identifier: doi.org/10.25921/11da-3x54 | Total Phosphorus Concentration ($\mu$g l$^{-1}$) | 111 |
| NOAA GLERL Western Lake Erie Sampling | Digital Object Identifier: doi.org/10.25921/11da-3x54 | Total Phosphorus Concentration ($\mu$g l$^{-1}$) | 1145 |
| National Center for Water Quality Research at Heidelberg University | https://ncwqr-data.org/ | Total Phosphorus Concentration ($\mu$g l$^{-1}$) | 2258 |
| NOAA Great Lakes Coastal Forecasting System | https://www.glerl.noaa.gov/res/glcfs/ | Surface Currents (m North, m East) | 1020318 |

**Appendix B**

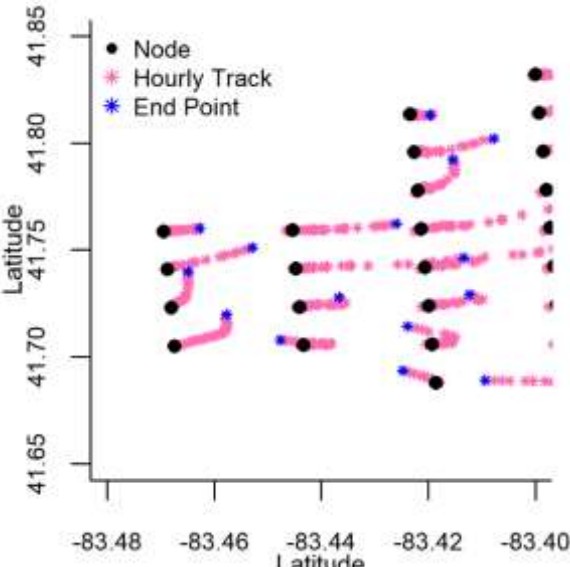

Figure B1. Surface current data was available hourly within western Lake Erie. These 24-hour data were used to track the daily
movement of water from each node.

**Appendix C**

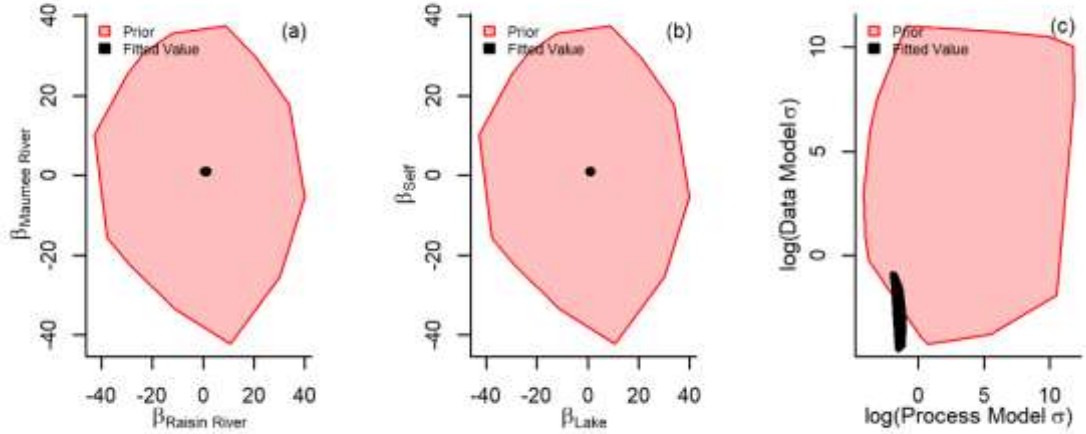

Figure C1. Non-informative priors were used to fit the state-space coefficients of the Maumee River ($\beta_{mau}$, a. all values represented as red polygon), River Raisin ($\beta_{ras}$), the western Lake Erie nodes subject to movement ($\beta_{lake}$, b. red polygon), and those Lake Erie nodes which did not encounter sufficient water movement to associate to an "upstream" node ($\beta_{self}$). The fitted values for every year (a. and b., all values represented as black polygon) do not appear to be overly influenced by the uninformed priors. The log data model and process model uncertainty (c) for every year also were well identified (the uncertainty σ was logged to aid in visually comparing prior and fitted values).

Table C1. State-space models in western Lake Erie fit coefficients predicting TP concentrations from the previous time-step within the Maumee
River, the River Raisin, Lake Erie, and lake locations where the current time-step is informed by the same location, $\beta_{mau}$, $\beta_{rai}$, $\beta_{lake}$, and $\beta_{self}$, respectively. The 95% predictive interval (PI) for each year and coefficient was examined. The same models fit annual process model and data model precision.

| Year | $\beta_{mau}$ | | $\beta_{rai}$ | | $\beta_{lake}$ | | $\beta_{self}$ | | Process σ | | Data σ | |
|------|------|------|------|------|------|------|------|------|------|------|------|------|
| | 95% PI | | 95% PI | | 95% PI | | 95% PI | | 95% PI | | 95% PI | |
| 2008 | 0.989 | 1.008 | 0.985 | 1.013 | 0.986 | 0.999 | 0.963 | 1.012 | 0.263 | 0.331 | 0.02 | 0.098 |
| 2009 | 0.993 | 1.005 | 0.992 | 1.007 | 0.991 | 1.000 | 0.984 | 1.002 | 0.168 | 0.193 | 0.125 | 0.165 |
| 2010 | 0.994 | 1.003 | 0.993 | 1.006 | 0.988 | 0.994 | 0.992 | 1.001 | 0.13 | 0.183 | 0.326 | 0.404 |
| 2011 | 0.993 | 1.006 | 0.99 | 1.008 | 0.991 | 1.006 | 0.985 | 1.007 | 0.174 | 0.209 | 0.017 | 0.061 |
| 2012 | 0.993 | 1.007 | 0.804 | 1.186 | 0.997 | 1.008 | 0.983 | 1.005 | 0.177 | 0.218 | 0.017 | 0.062 |
| 2013 | 0.993 | 1.007 | 0.992 | 1.008 | 0.992 | 1.003 | 0.978 | 1.008 | 0.181 | 0.277 | 0.135 | 0.25 |
| 2014 | 0.992 | 1.005 | 0.99 | 1.006 | 0.991 | 1.001 | 0.991 | 1.018 | 0.173 | 0.235 | 0.145 | 0.232 |
| 2015 | 0.992 | 1.006 | 0.989 | 1.007 | 0.992 | 1.003 | 0.982 | 1.003 | 0.212 | 0.264 | 0.156 | 0.238 |
| 2016 | 0.993 | 1.005 | 0.991 | 1.005 | 0.996 | 1.004 | 0.983 | 1.001 | 0.161 | 0.2 | 0.146 | 0.214 |
| 2017 | 0.993 | 1.006 | 0.991 | 1.008 | 0.995 | 1.003 | 0.978 | 0.998 | 0.197 | 0.24 | 0.019 | 0.079 |
| 2018 | 0.992 | 1.007 | 0.99 | 1.008 | 0.983 | 0.991 | 0.966 | 0.997 | 0.221 | 0.26 | 0.018 | 0.065 |

**Appendix D**

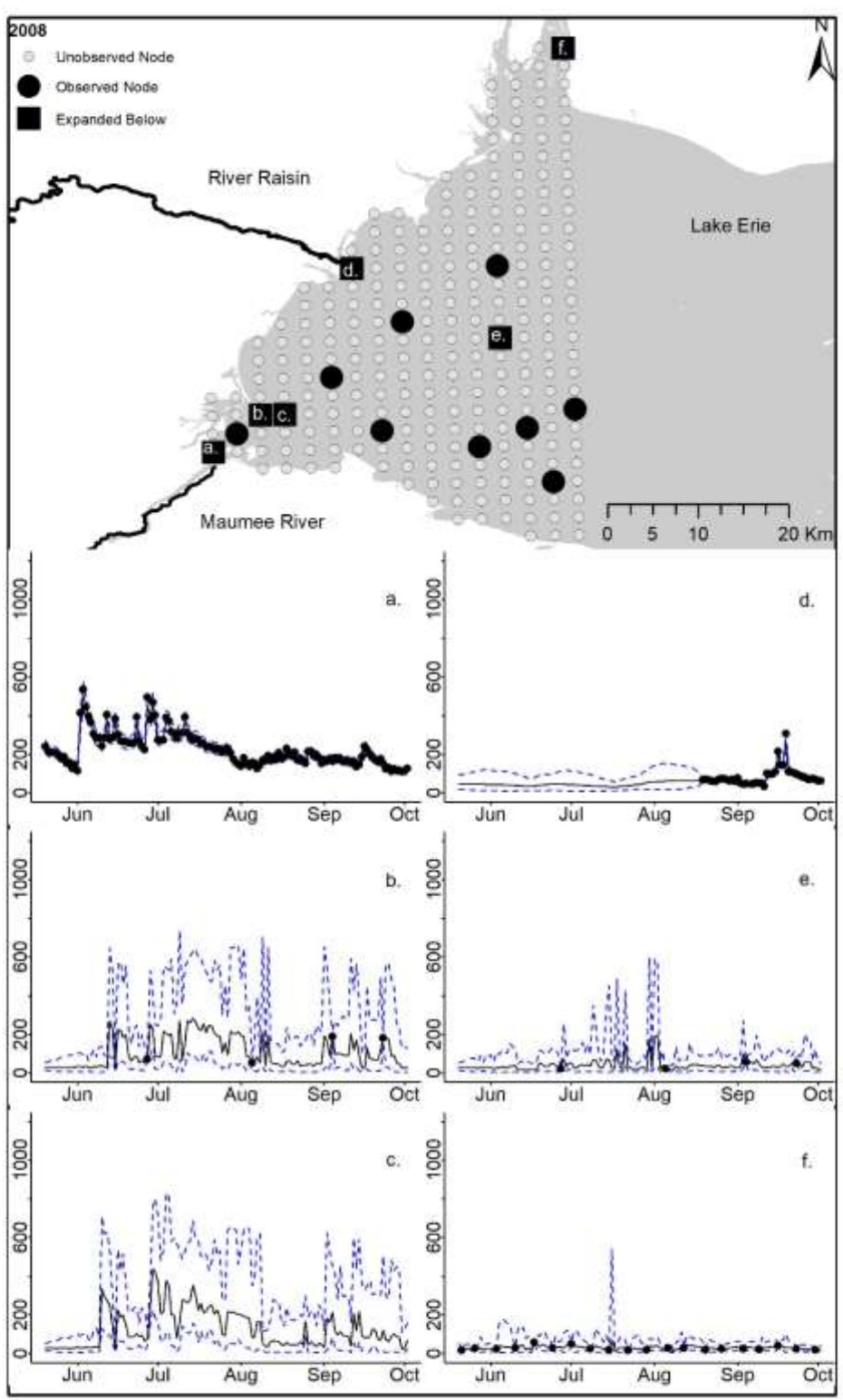

**Figure D1. For 2008 the total phosphorus concentration (µg l⁻¹) at observed and unobserved nodes were estimated from the model posterior samples. Mean (solid black line) and 95% PI (dashed blue line) for the model posterior samples of each node at every day for (a) the Maumee River, (b,c,e,f) western Lake Erie nodes, and (d) the River Raisin.**

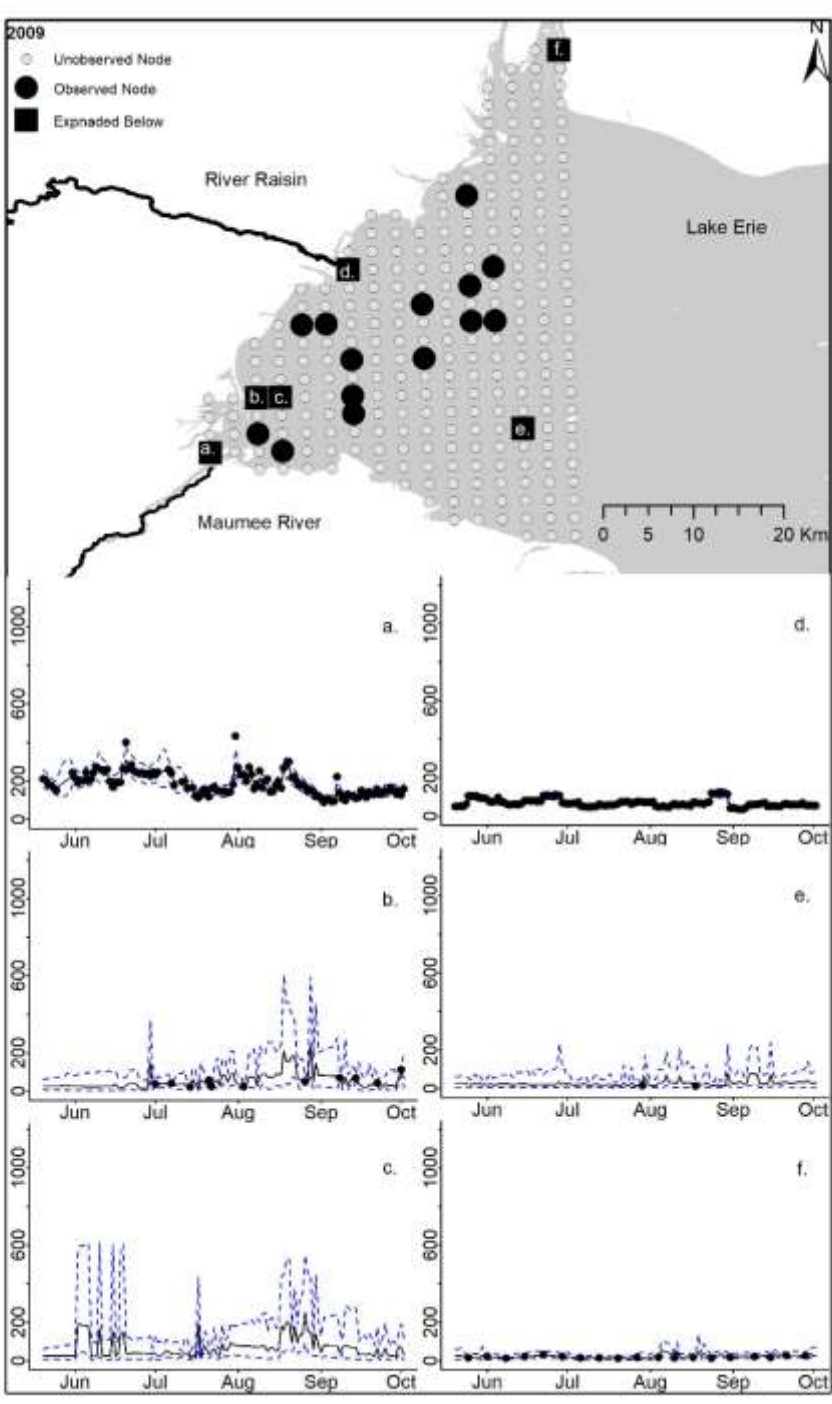

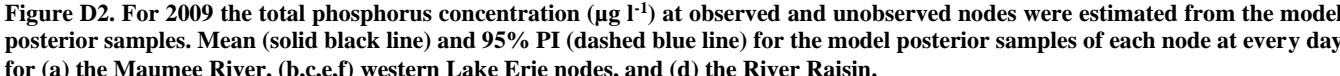

**Figure D2. For 2009 the total phosphorus concentration (µg l⁻¹) at observed and unobserved nodes were estimated from the model posterior samples. Mean (solid black line) and 95% PI (dashed blue line) for the model posterior samples of each node at every day for (a) the Maumee River, (b,c,e,f) western Lake Erie nodes, and (d) the River Raisin.**

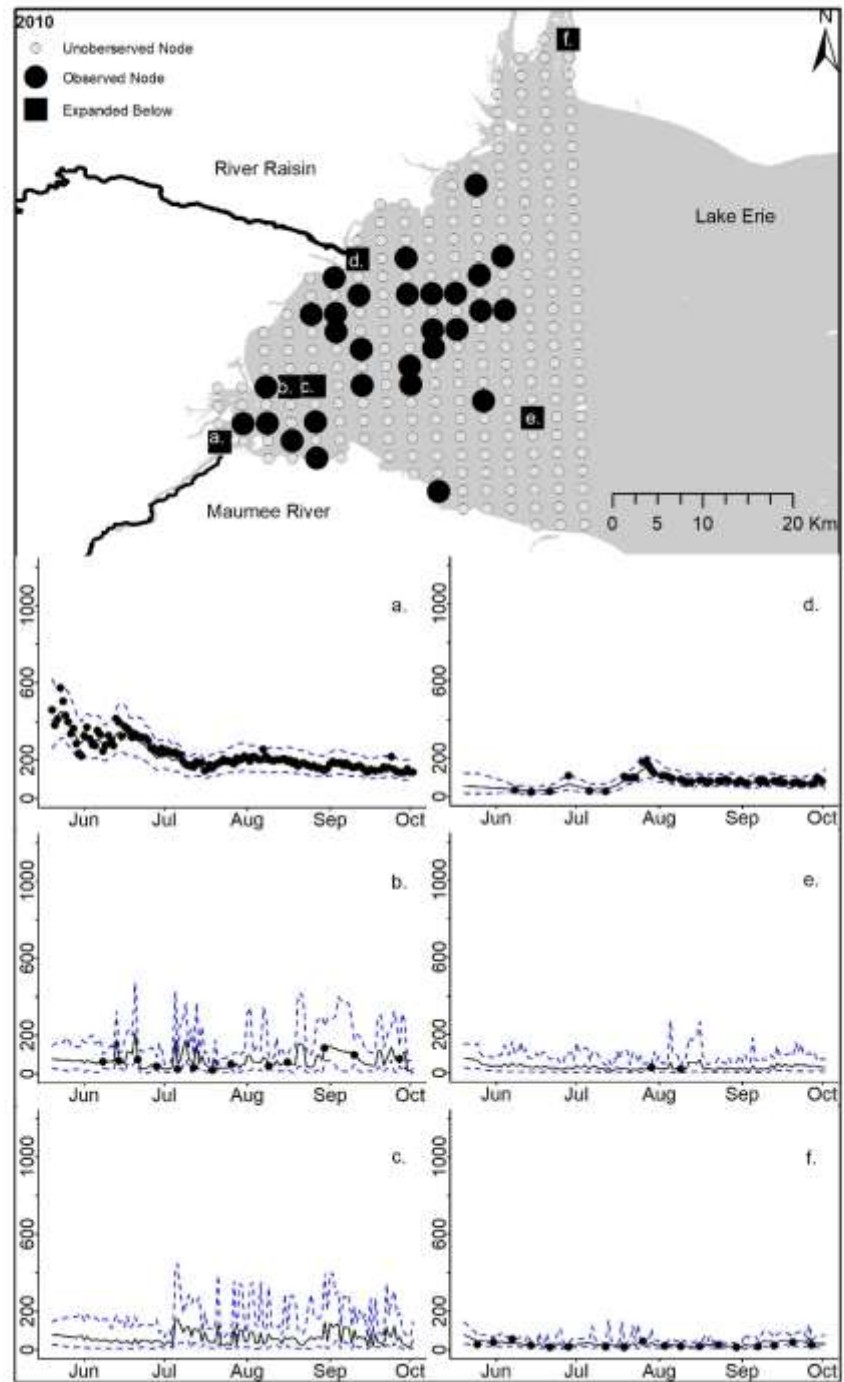

**Figure D3. For 2010 the total phosphorus concentration (µg l$^{-1}$) at observed and unobserved nodes were estimated from the model posterior samples. Mean (solid black line) and 95% PI (dashed blue line) for the model posterior samples of each node at every day for (a) the Maumee River, (b,c,e,f) western Lake Erie nodes, and (d) the River Raisin.**

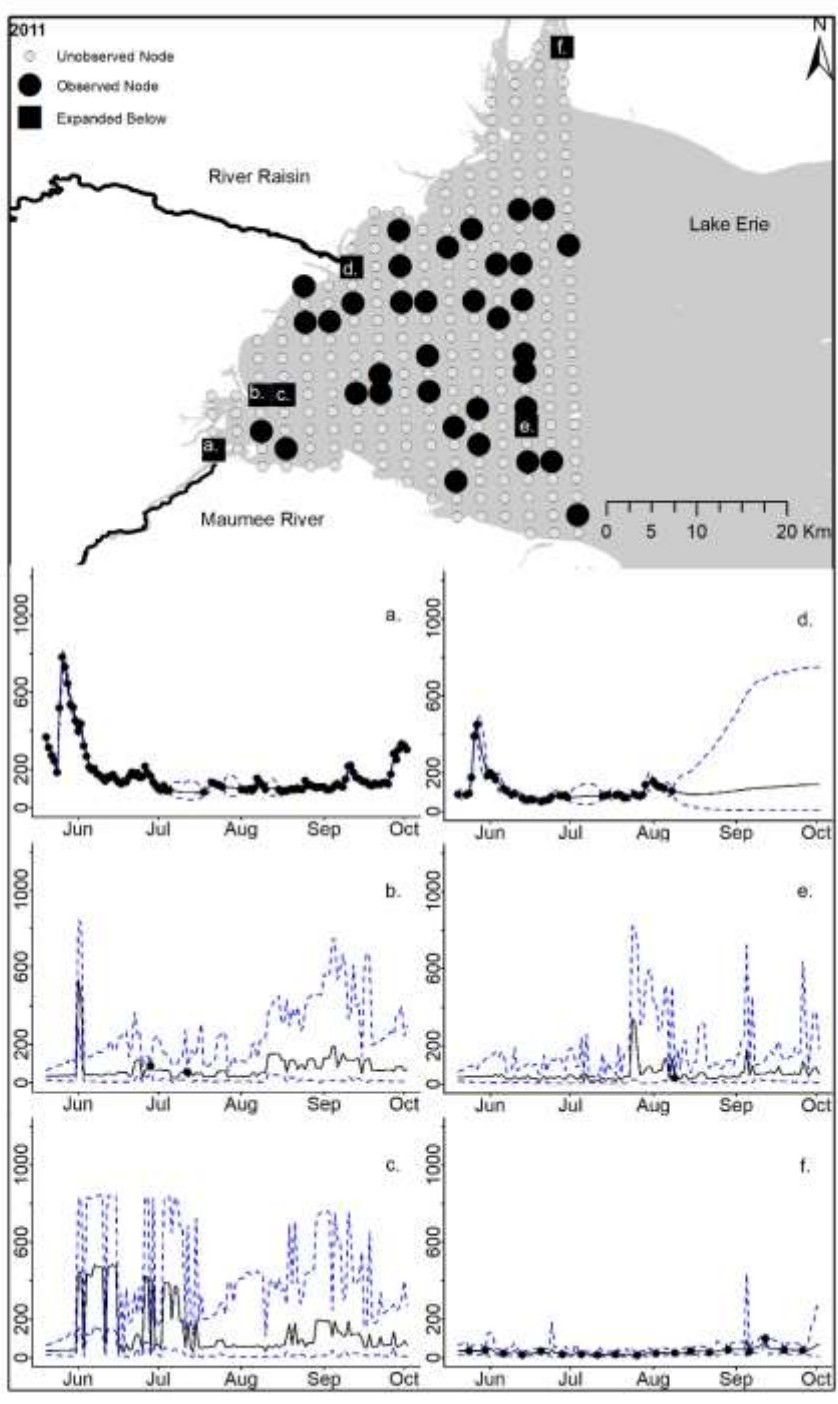

**Figure D4. For 2011 the total phosphorus concentration (µg l⁻¹) at observed and unobserved nodes were estimated from the model posterior samples. Mean (solid black line) and 95% PI (dashed blue line) for the model posterior samples of each node at every day for (a) the Maumee River, (b,c,e,f) western Lake Erie nodes, and (d) the River Raisin.**

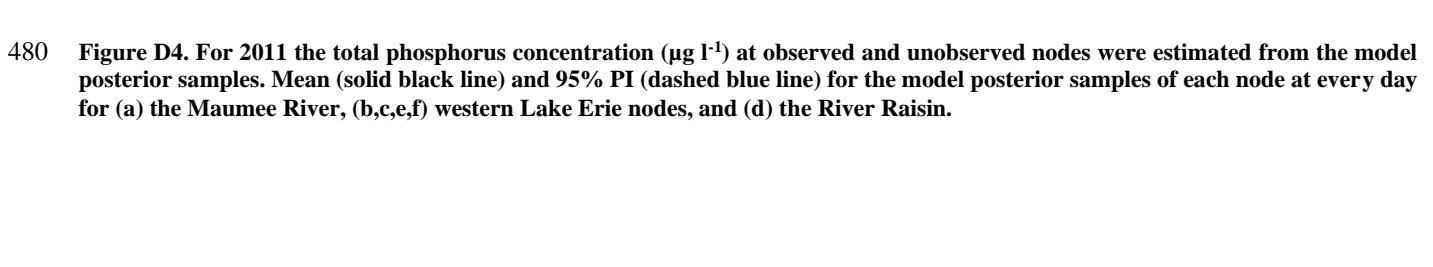

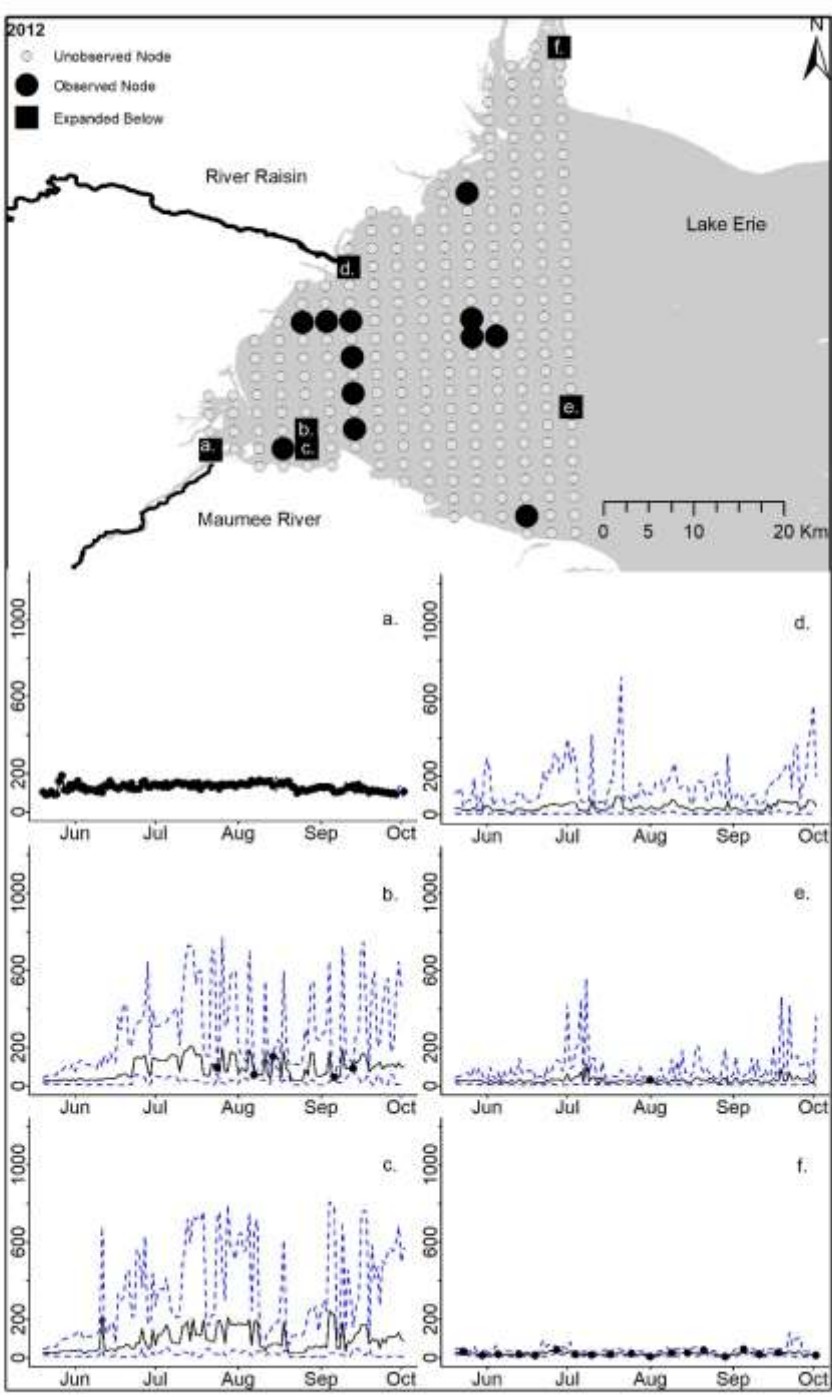

**Figure D5.** For 2012 the total phosphorus concentration (µg $l^{-1}$) at observed and unobserved nodes were estimated from the model posterior samples. Mean (solid black line) and 95% PI (dashed blue line) for the model posterior samples of each node at every day for (a) the Maumee River, (b,c,e,f) western Lake Erie nodes, and (d) the River Raisin.

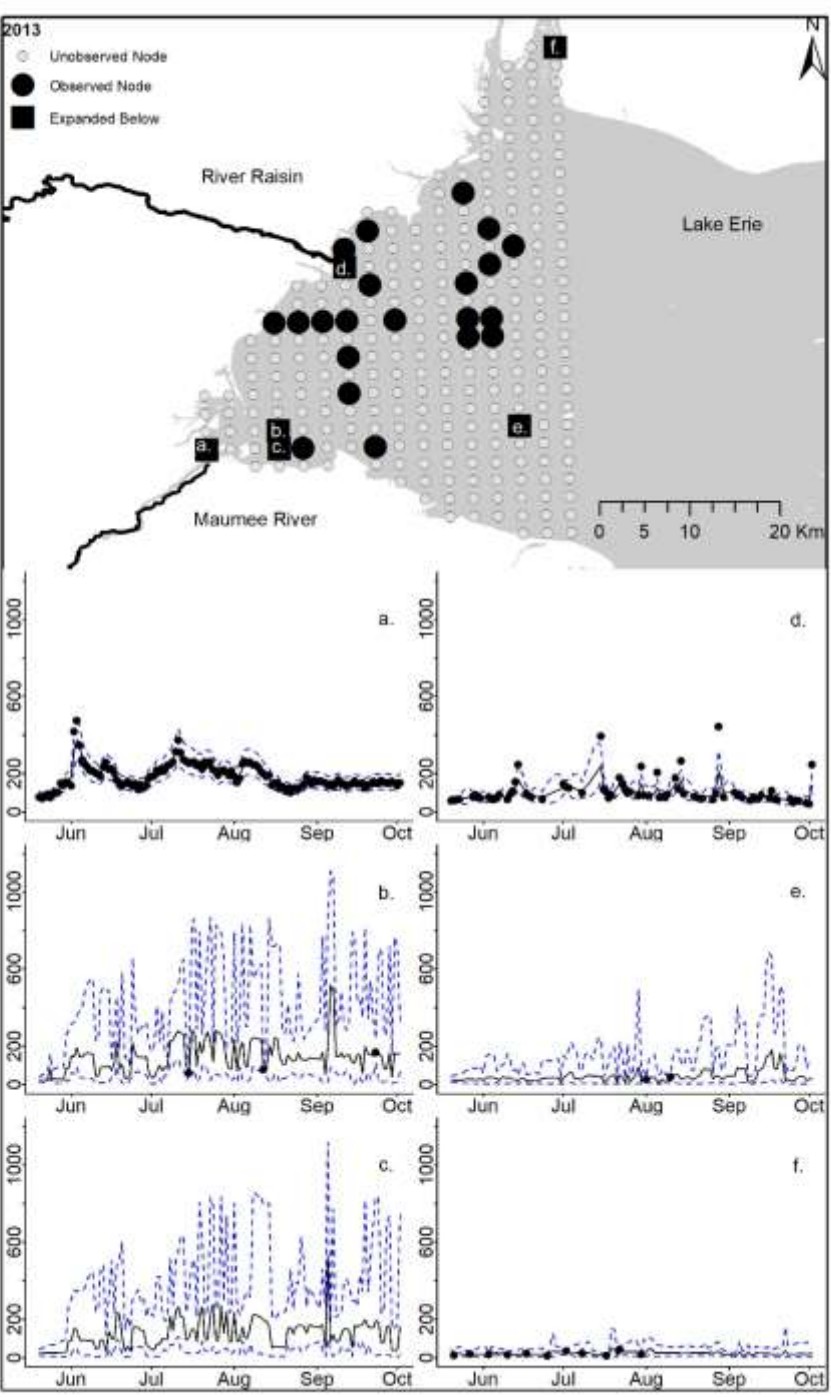

**Figure D6. For 2013 the total phosphorus concentration (μg l$^{-1}$) at observed and unobserved nodes were estimated from the model posterior samples. Mean (solid black line) and 95% PI (dashed blue line) for the model posterior samples of each node at every day for (a) the Maumee River, (b,c,e,f) western Lake Erie nodes, and (d) the River Raisin.**

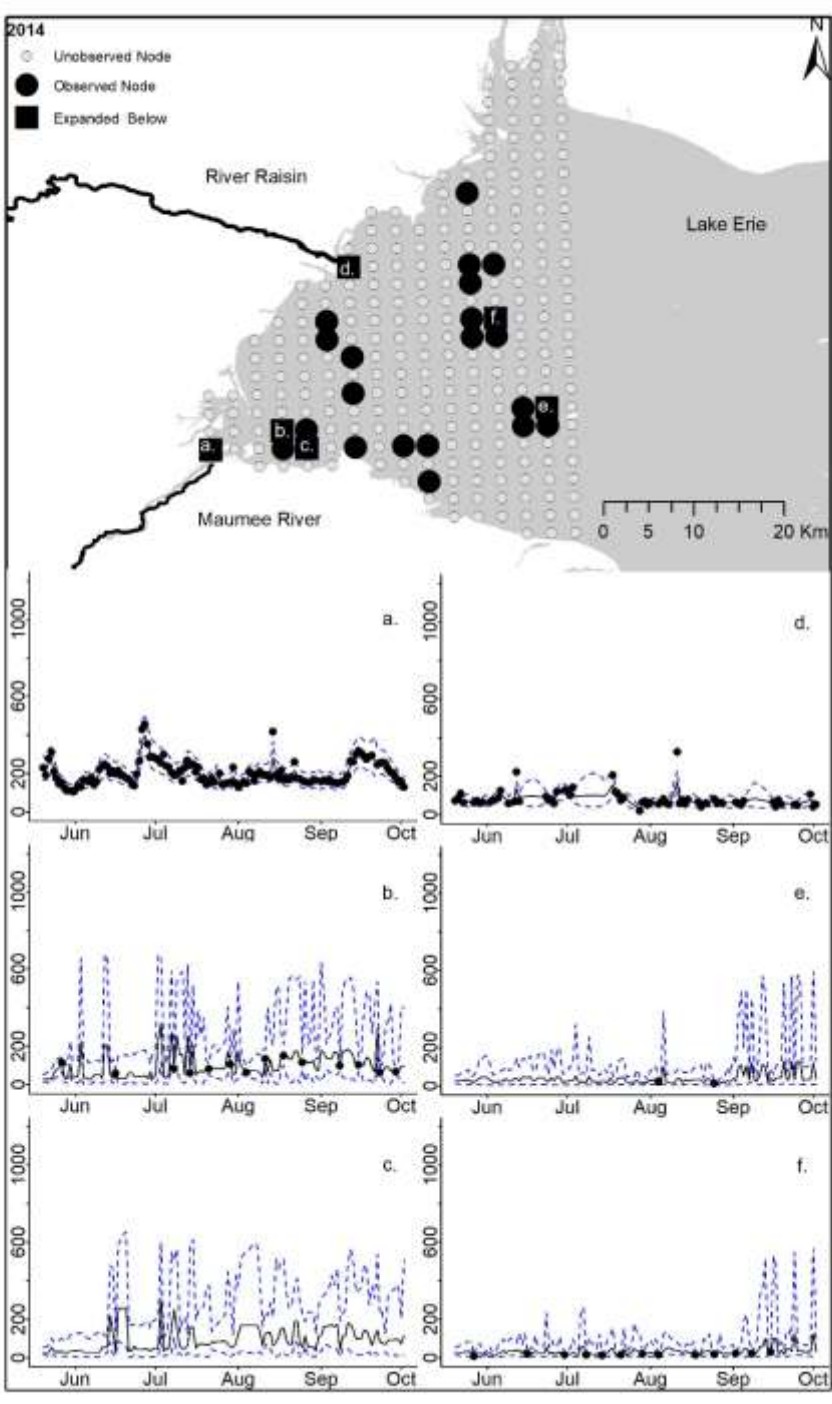

**Figure D7. For 2014 the total phosphorus concentration (μg l$^{-1}$) at observed and unobserved nodes were estimated from the model posterior samples. Mean (solid black line) and 95% PI (dashed blue line) for the model posterior samples of each node at every day for (a) the Maumee River, (b,c,e,f) western Lake Erie nodes, and (d) the River Raisin.**

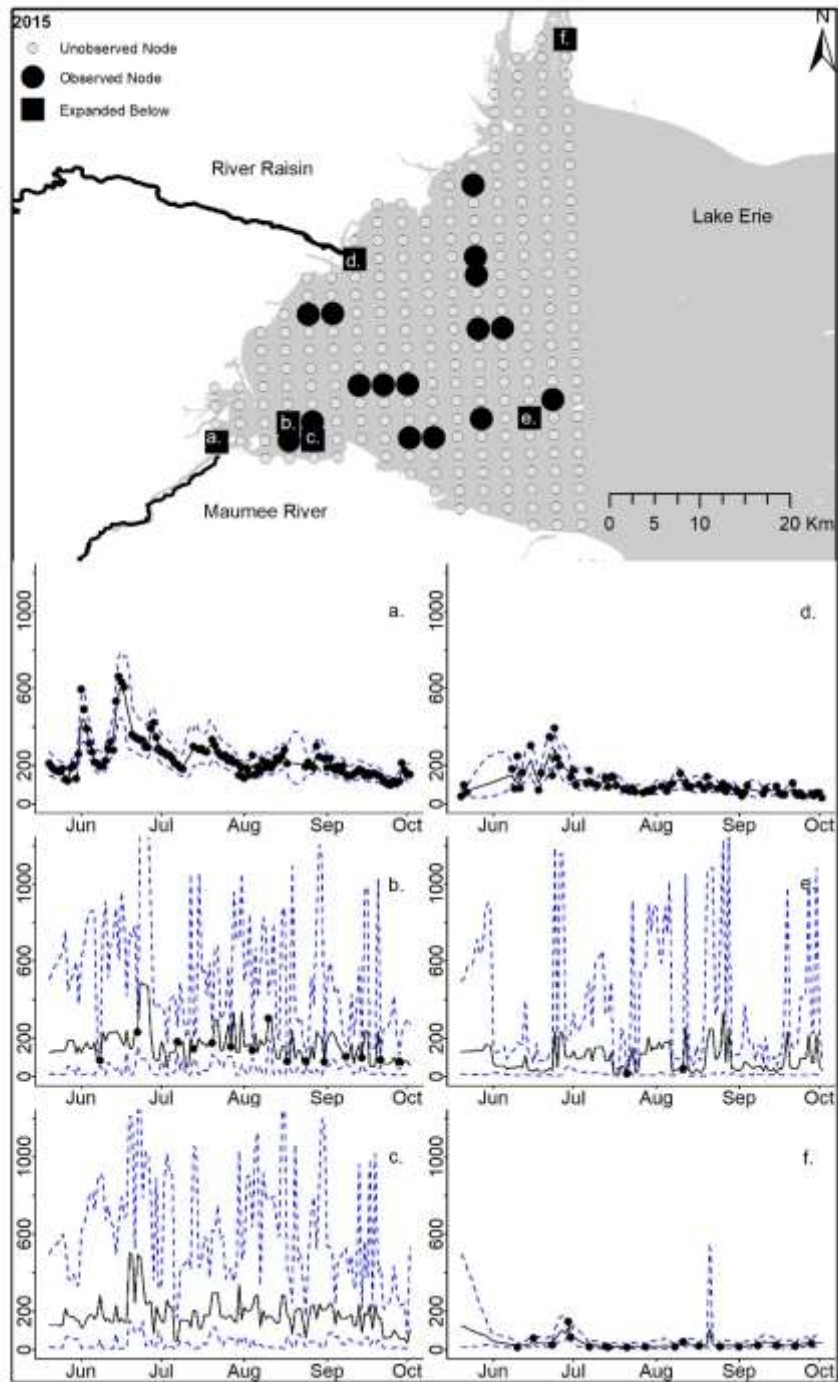

**Figure D8.** For 2015 the total phosphorus concentration (μg l$^{-1}$) at observed and unobserved nodes were estimated from the model posterior samples. Mean (solid black line) and 95% PI (dashed blue line) for the model posterior samples of each node at every day for (a) the Maumee River, (b,c,e,f) western Lake Erie nodes, and (d) the River Raisin.

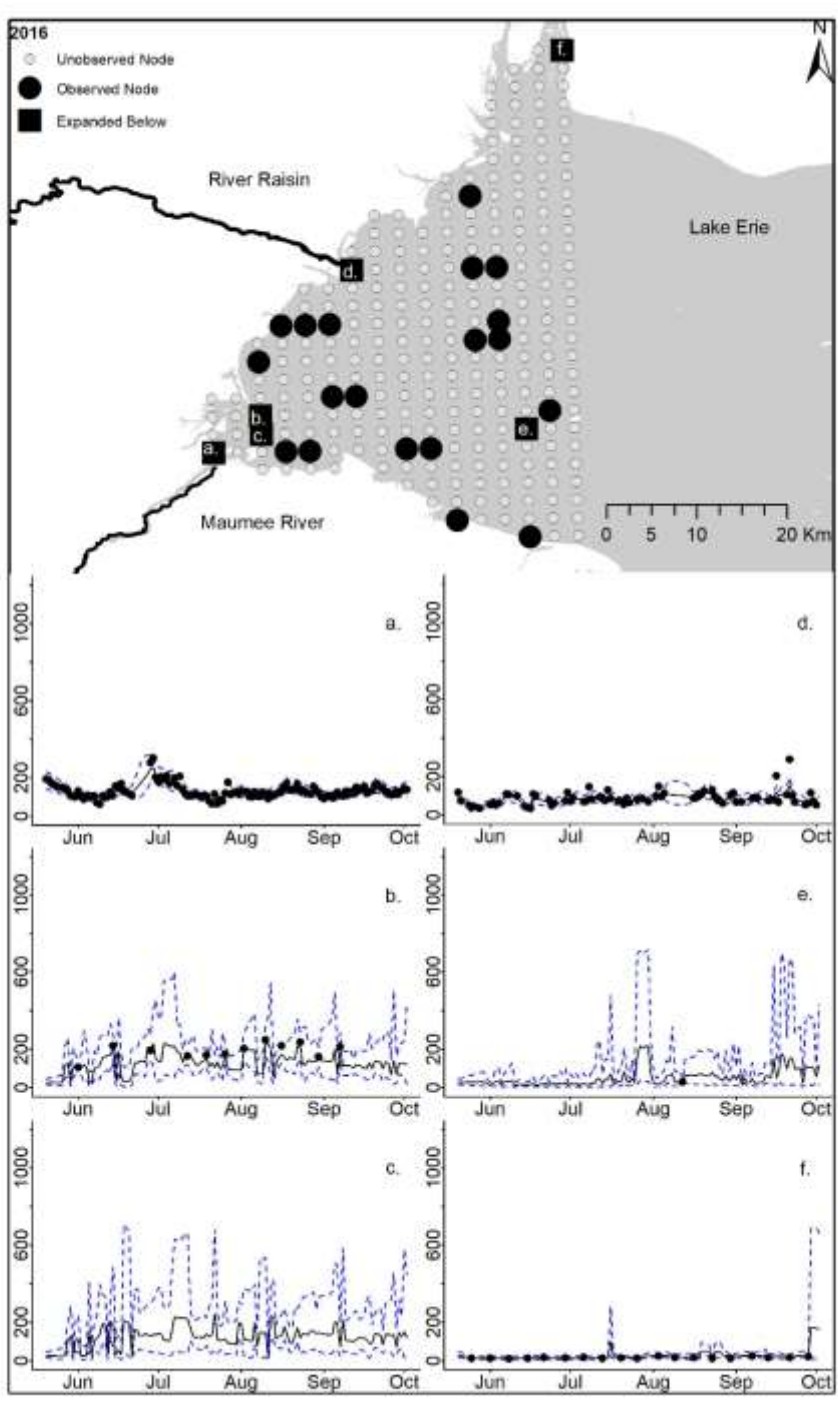

**Figure D9. For 2016 the total phosphorus concentration (µg l$^{-1}$) at observed and unobserved nodes were estimated from the model posterior samples. Mean (solid black line) and 95% PI (dashed blue line) for the model posterior samples of each node at every day for (a) the Maumee River, (b,c,e,f) western Lake Erie nodes, and (d) the River Raisin.**

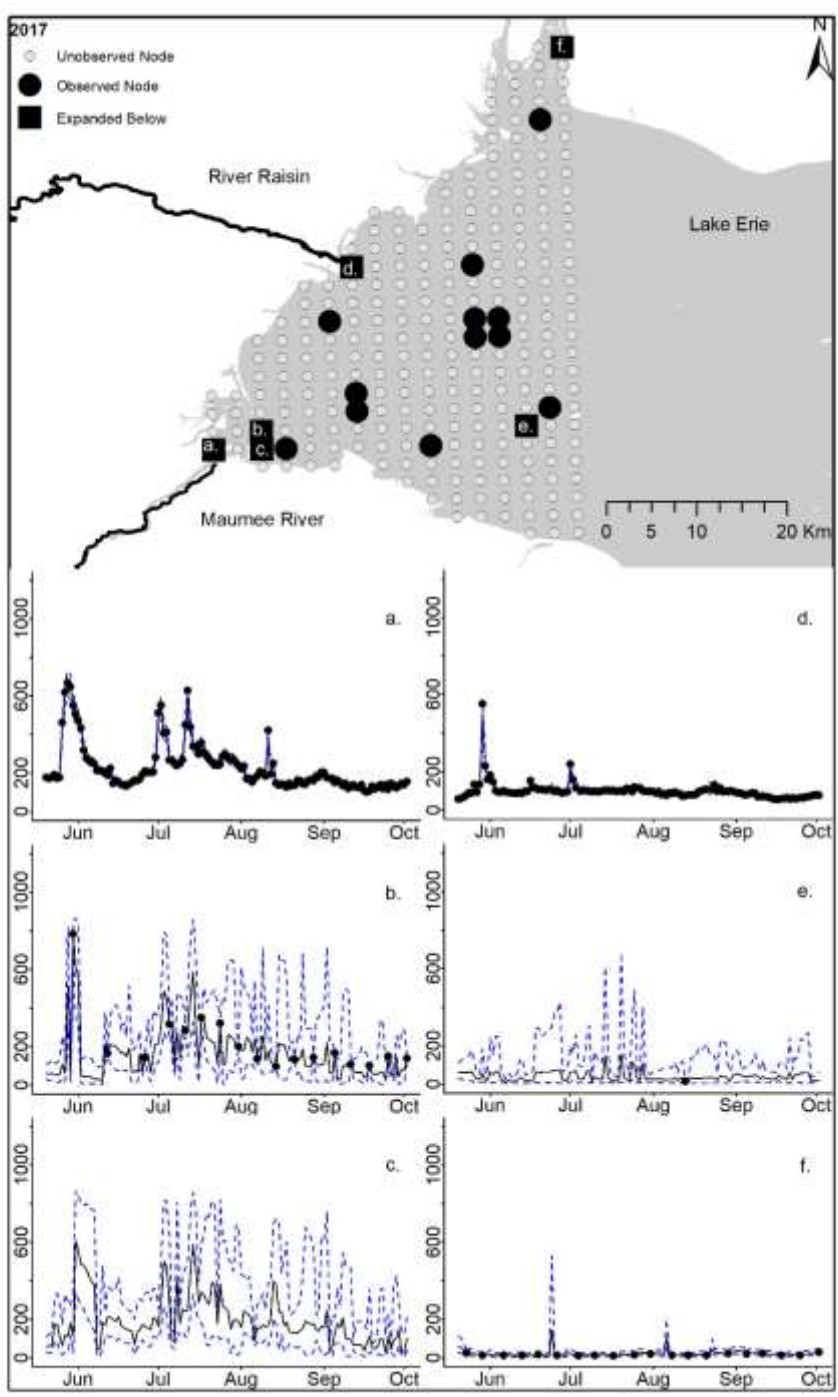

**Figure D10. For 2017 the total phosphorus concentration (µg l⁻¹) at observed and unobserved nodes were estimated from the model posterior samples. Mean (solid black line) and 95% PI (dashed blue line) for the model posterior samples of each node at every day for (a) the Maumee River, (b,c,e,f) western Lake Erie nodes, and (d) the River Raisin.**

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
