# Peer review of "Spatially Referenced Bayesian State-Space Model of Total Phosphorus in western Lake Erie"

_Hydrology and Earth System Sciences, 2021_

## Referee Comment (RC2)

Review of:

**Spatially Referenced Bayesian State-Space Model of Total Phosphorus in western Lake Erie**

July 16, 2021

Referee: Ken Newman, Biomathematics & Statistics Scotland, and School of Mathematics, University of Edinburgh

**1   Overall comments**

Developing a quantitative, explanatory model that links total phosphorus (TP) loads in rivers that discharge into Lake Erie to the TP loads in the lake is useful for both increasing scientific understanding of the processes and for assessing potential management actions. The compilation and "wrangling" of both the lake and river TP measurements and the surface current data was no small task. The state-space model (SSM) framework seems quite appropriate given the time series nature of the data, and the spatial structure of the process model for these data is a crucial feature. The subsequent use of the fitted SSM to conduct "what if" exercises on changes in riverine TP loads shows the work's *potential* utility as a decision support tool.

I have major concerns about the formulation and fitting of the SSM and the explanation of the results.

1. Adjacency matrix: instead of calculating mean daily eastward and northward water velocity (m/sec) for a given node, using the hourly measurements at that node, why not follow the hour by hour trajectory of points across a day? On a per cell basis at hour 1 start with a point at the cell center (the node), say $(Lon_{c,1}, Lat_{c,1})$ and use eq'ns 3 and 4 to advance that point to its longitude and latitude at hour 2, $(Lon_{c,2}, Lat_{c,2})$. Then for whatever cell that point is in, apply eqn's 3 and 4 again to move it to $(Lon_{c,3}, Lat_{c,3})$, and so on until reaching $(Lon_{c,24}, Lat_{c,24})$.

2. A truncated normal distribution for the process model does not make sense. The state component is the "true" log TP concentration and that is not a truncated value. A truncated normal could be used for an observation model, however.

3. Given that year-specific SSMs were fit, it would be useful to compare the posterior distributions for the four process model parameters, $\beta_{mau}$, $\beta_{rai}$, $\beta_{self}$, and $\beta_{lake}$. At a minimum there needs to be summaries about the posteriors of these parameters, and some discussion and interpretation of the values are needed.

4. Related to item 3, a more comprehensive approach to fitting these data would be a hierarchical SSM. Model the slope coefficients (the $\beta$'s) as random variables coming from a generating distribution that reflects between year (or environmental) variation; eg., $\beta_{mau,y} \sim \text{Normal}(\mu_{\beta_{mau}}, \sigma_{\beta_{mau}})$.

5. Nothing has been said about the estimates of the process model precision ($Q$) and the obs'n model precision ($R$). How do they vary between years? There are often weak identifiability problems with these kinds of linear Gaussian SSMs (see Auger-Méthé et al. (2016)).

6. Regarding application of the SSM to assess the effect of reducing TP loads in the Maumee River, assuming year-specific SSMs were fit, then the particular SSM that was used needs to be stated. Also related to 3, it would be good to report results for each of the year-specific SSMs to show (some of) the uncertainty in the assessments.

7. While it is good to have included the `R` code and data for fitting the SSM to the 2018 data, something needs to be said about computational time. I ran the code, but it had not completed after 14 hours, and I could not therefore verify any results.

**2    Detailed technical comments**

p=page, L=line.

1. Introduction, p3, L70: Why test a hypothesis of linearity? A more general aim would be to quantify the nature of a relationship, be it linear or nonlinear.

2. Methods

   (a) p3, L73-79. Is there overlap in the sampling on Lake Erie? For example, do two agencies collect data on the same cells?

   (b) p3, L81. By station is that just referring to the rivers, or does that include the lake? When multiple samples were collected from a "station on a single day, how much variation was there in the measure value? Such information could be used in the observation model of the SSM.

   (c) pp 3-4, L84-100. It would help to create an example figure showing the geometry (or trigonometry) underlying the calculation of a particle's change in position at time $t$ to time $t+1$. For example, draw an x-y plot with points $p_t$ and $p_{t+1}$ with coordinates x=longitude and y=latitude. Draw a right triangle, where the vertical and horizontal sides are parallel to the y-axis and x-axis respectively, and the hypotenuse connects $p_t$ and $p_{t-1}$. So the vertical side indicates the "northerly" movement, the horizontal side indicates "easterly" movement. (For what it's worth, I find thinking of this as a step direction and step size process—as is done in animal movement modeling.)

   (d) p3, L89. A refinement would be fit a spatially smooth velocity map rather than use values at "nodes", though I'm not sure how much practical effect that would have.

   (e) p4, L101. Does anything need to be handled differently at cells on the perimeter? Such cells could be transferring TP to cells outside the spatial window, presumably.

   (f) p4, L109. References to SSM literature are strongly recommended; e.g., Durbin and Koopman (2012); Shumway and Stoffer (2019). Discussion of the distinction between inference about latent states and fixed parameters would be good.

   (g) p4, L119. Say something about the $\beta$'s and what they mean. For example, regarding $\beta_{self}$ and $\beta_{lake}$, are they likely less than 1? Is there some loss of TP between a source point $k$ and an end point $n$ from one day to the next? Is any sort of seasonality expected in the river values? This would make the assumption of constant $\beta_{mau}$ and $\beta_{rai}$ suspect.

   (h) p5 L126-128: The joint prior described for $\beta_{mau}$ and $\beta_{rai}$ does not exactly match what the `R` code indicates:

$$\mu_{\beta_{river}} \sim \text{Normal}(0, 1/\sqrt{0.01})$$
$$\tau_{\beta_{mau}} \sim \text{Gamma}(0.001, 0.001)$$
$$\tau_{\beta_{rai}} \sim \text{Gamma}(0.001, 0.001)$$
$$\beta_{mau} \sim \text{Normal}\left(\mu_{\beta_{river}}, 1/\sqrt{\tau_{\beta_{mau}}}\right)$$
$$\beta_{rai} \sim \text{Normal}\left(\mu_{\beta_{river}}, 1/\sqrt{\tau_{\beta_{rai}}}\right)$$

   where the 2nd parameter in the normal is the standard deviation.

   (i) p5, L137-138: Why is cross-validation needed to make comparisons of goodness of fit across years? Doesn't the Bayesian $R^2$ do that? Cross-validation is more often used for model selection, which is the focus of both the referenced Vehtari et al and the Piironen and Vehtari papers.

   (j) p6,L155-164: I cannot tell what is being done here. Is this necessary?

   (k) p6, L171-174: Write down an equation for deflection, $d_{n,y}$, and for the normalized estimate.

(l) p6, L179-185: Are 252 regressions being fit each year? Write down what effective load, $\tilde{l}_{n,y}$ means—is it an average? Why is there a subscript $n$ if it is measured on the Maumee? I'd be curious about identifiability issues/posterior correlations with the parameters in eq'n 11, too— seems overly complicated. Would it make sense to take average "raw" $y$ and take the log of that?

(m) R code: The priors for precision to the Lake, Maumee River, and River Raisin are calculated as the inverse of standard deviation; shouldn't that be the variance? Also using the data to set priors is questionable.

(n) Data for R code: why does the adjacency object `use` have 4 values for the "next" cell, at time $t + 1$? Only the first is used in the code.

(o) Sensitivity analysis for the priors needs to be conducted.

3. Results

(a) p7, around L185. Say something about the ranges and averages of northerly and easterly velocities (m/sec), of TP concentrations (distinguishing between Maumee River, River Raisin, and western Lake Erie), and of calculated distances moved in a single day.

(b) p7, L195-199. In addition to the summaries in Table 1, show some plots of posterior mean values for some of the cells across the 136 day period against corresponding observations. Show an example histogram (or two) of the distribution of predicted log concentrations with the observation. Presumably these $R^2$ are only calculated on the cells with observations. I don't understand what the cross-validated measure is doing (relates to earlier comment in methods).

(c) p7, L200-205. As mentioned above need to report out results on parameter estimates for the $\beta$ (each of the 4 for all 11 years) and report out the process and observation model standard deviations ($1/\sqrt{Q}$ and $1/\sqrt{R}$, based on `R` code).

**Minor editorial remarks**

1. Throughout, consider using the word "cell" instead of "node" as a node is usually interpreted as a point. Node could then refer to cell center.

2. Section 2 Methods.

(a) pp 3-4. Consider creating a new subsection for the material in the first paragraph (L84-107) of the Model Description subsection, maybe naming it Construction of an Adjacency matrix.

(b) p4, L109. Emphasize that 11 different SSMs will be fit.

(c) p4, L119. The observations are *modeled* with a normal distribution; they are not *estimated* with a normal. As mentioned previously, a truncated normal (perhaps just on the left) could be used to account for measurement limitations.

(d) p5, L139. Consider renaming Section 2.2.2 "Fitting the SSM" as "SSM Fit" could be interpreted as a result not a method.

(e) p5,L133: Perhaps move text beginning with "The model was run.." into Section 2.2.2. Could delete/move material in the sentence beginning "The efficacy of.." as it is redundant with material on L140-151.

(f) p5, L136. "efficacy" seems an odd choice, why not goodness of fit?

(g) p5,L137: The Vehtari, et al., 2017 paper does not refer to Bayesian $R^2$. Give a mathematical definition of the $R^2$ here: not clear to me what resolved and residual variances mean.

(h) p5,L138 What does "utility" mean? And what value of $K$ was used?

(i) p5,L145: Cross-validation is meant for CV not coefficient of variation?

(j) p5, L154: What does preferentially mean?

(k) p6, L165: Instead of Model Experimentation, wouldn't Model Usage or Application make more sense here (and elsewhere)?

(l) 6, L166: As said previously, the SSM for which year was used?

3. Section 3 Results.

(a) p7, L187-193. Re: the degree of missing data, I think it would be easier to follow by first saying how many space-time cells there are and then give the number with data: "For the Lake, there 252*11*136 = 376,992 cells of which 1218 had data, and for the two rivers, there were 2*11*136 = 2992 cells, of which 2258 had observations". Save the discussion of the inference for cells without data till later.

(b) p7, around L188. It would help to see an example plot that shows the spatial dist'n of cells, in a given year, that had at least one observation (see Figure 1).

(c) Also a plot showing "source" cells and "end" cells would be good to show the adjacency. (Note: in the R code, `use` object has 4 values in the second dimension, but it appears that only the 1st value is used as the adjacency matrix—what are the other 3 values for?)

(d) p7, around L188. Add a plot or two of the log(TP) concentrations. See Figure 2.

(e) p7, L195: "efficiency" here, but "efficacy" in Methods, but would model quality or goodness-of-fit be more appropriate?

(f) p7, L197: typo: Table 1 not Table 2.

(g) p7, L202: Say that Figure 2 shows 2018 and Figure B.1 in Appendix B shows 2006. The captions in those figures need to indicate that the black dots are observed values.

4. p8,L222 (and p10, L294): "amending" seems an odd choice: to amend would mean to modify data in such a way that the modified data are an improvement.

5. p8, L231: Was the notation $k$ for the derived adjacency matrix used before? $k$ was just the "source" cell from time $t$ which feed a "sink" cell at time $t + 1$?

**References**

Auger-Méthé, M., Field, C., Albertsen, C. M., Derocher, A. E., Lewis, M. A., Jonsen, I. D., and Flemming, J. M. (2016). State-space models' dirty little secrets: even simple linear gaussian models can have estimation problems. *Scientific reports*, 6(1):1–10.

Durbin, J. and Koopman, S. J. (2012). *Time series analysis by state space methods*. Oxford university press.

Shumway, R. and Stoffer, D. (2019). *Time series: a data analysis approach using R*. CRC Press.

[Figure]

Figure 1: Cells with any TP data in 2018 marked in blue.

[Figure]

Figure 2: TP data for 2018 by cell.

---

## Author Comment (AC1)

**Reviewer Comment**

**Author Response**

The authors applied a spatially referenced Bayesian state-space model to conduct an analysis on how in-lake (Erie) TP concentrations are affected by the adjacent river basin. In general, the paper is adequately written (this means that the authors could improve the manuscript to succeed a smoother experience for the reader) and I find their effort decent in terms of methodological analysis. The presentation of the reults is satisfactory, with convincing explanations when needed.

GENERAL COMMENTS:

**R1C1**  Although that in introduction you speak about the absence of scientific investigation in correlating out-of-the lake and in-lake nutrient concentrations, I think that the authors must highlight their contribution better. For this reason, I propose to add some lines explaining your contribution to the methodology used and how this progresses science beyond the state-of-the-art .

The authors propose adding text and an appropriate citation in **Section 1.**

> "Our contribution to the ecological state space model methodology is incorporating a surface current derived adjacency matrix, combined disparate agency field data, and incorporate data from rivers as sources of information in fitting estimated values within a system dominated by missing values. Together our contribution will fit values in the absence of observations and allow experimentation in archival data previously not possible."

**R1C2**  By applying the Bayesian methodology and using three different chains, in my opinion, you should speak about convergence. It is a prerequisite for the model used here.

We agree with this reviewer comment, thank you for pointing out the omission. The authors propose adding text and an appropriate citation in **Section 2.2.1** where we describe fitting the models.

> "The chain convergence was monitored by Gelman and Rubin's convergence diagnostic. Scale factors less than 1.1 were used to define when chains had converged (Plummer, 2019)."

LINE-SPECIFIC COMMENTS:

**R1C3**  Line 32: I suggest to add one recent and very relevant reference after Schneider and Blaha, 2020: Mellios, N. K., Moe, S. J. and Laspidou, C.: Using Bayesian hierarchical modelling to capture cyanobacteria dynamics in Northern European lakes, Water Research, 186, 116356.

The authors will change the manuscript as suggested by the reviewer.

**R1C4** Line 37: Please use 2 as superscript in km2. Correct t throughout the manuscript.

The authors will change the manuscript as suggested by the reviewer.

**R1C5** Line 65: My opinion is to erase "within".

The authors will change the manuscript as suggested by the reviewer.

**R1C6** Line 66: Please use "it does not" instead of "it is doeas not".

Our copy of the original manuscript does not contain this error, potentially an extra character was inserted during the PDF-upload-conversion process. We will ensure that the error is not reproduced.

**R1C7** Line 79: Please use "extend from" instead of "range from".

The authors will change the manuscript as suggested by the reviewer.

**R1C8** Line 111: Please erase the "the".

The authors will change the manuscript as suggested by the reviewer.

**R1C9** Line 188: Use "values" instead of "vales".

The authors will change the manuscript as suggested by the reviewer.

**R1C10** Line 190: After "were collected" use ";" instead of comma in order to make the sentence smoother for the reader.

The authors will change the manuscript as suggested by the reviewer.

**R1C11** Line 291: Please use "also" instead of "too".

The authors will change the manuscript as suggested by the reviewer.

---

## Author Comment (AC2)

**Reviewer Comment**

**Author Response**

Review of:
Spatially Referenced Bayesian State-Space Model of Total Phosphorus in western Lake Erie
July 16, 2021
Referee: Ken Newman, Biomathematics & Statistics Scotland, and School of Mathematics, University of Edinburgh
1 Overall comments
Developing a quantitative, explanatory model that links total phosphorus (TP) loads in rivers that discharge into Lake Erie to the TP loads in the lake is useful for both increasing scientific understanding of the processes and for assessing potential management actions. The compilation and \wrangling" of both the lake and river TP measurements and the surface current data was no small task. The state-space model (SSM) framework seems quite appropriate given the time series nature of the data, and the spatial structure of the process model for these data is a crucial feature. The subsequent use of the fitted SSM to conduct "what if" exercises on changes in riverine TP loads shows the work's potential utility as a decision support tool. I have major concerns about the formulation and fitting of the SSM and the explanation of the results.

**R2C1**  Adjacency matrix: instead of calculating mean daily eastward and northward water velocity (m/sec) for a given node, using the hourly measurements at that node, why not follow the hour by hour trajectory of points across a day? On a per cell basis at hour 1 start with a point at the cell center (the node), say (Lonc;1; Latc;1) and use eq'ns 3 and 4 to advance that point to its longitude and latitude at hour 2, (Lonc;2; Latc;2). Then for whatever cell that point is in, apply eqn's 3 and 4 again to move it to (Lonc;3; Latc;3), and so on until reaching (Lonc;24; Latc;24).

We changed our approach and remade the adjacency matrix as the reviewer suggested. The difference was very slight, e.g., in 2018 of the ~35,000 adjacencies only 5 were different using the suggested methodology. This suggests that the computationally intense hourly approach does not finely resolve information on water movement, at least in this location. The "hour-step" method, while not a deterministic particle-tracking model, did not improve performance in our system potentially because of our dominated west to east movement patterns. We used the reviewer's suggested method regardless because while our surface current data was insensitive to which method was used, differences between the two methods may be more extreme in other environments and thus the reviewers "hour-step" method would be more widely applicable.

We propose this text added to **Section 2.2** amending the adjacency matrix description.
"Hourly northward and eastward velocity (m day$^{-1}$) for each node for years 2008 to 2018 defined surface current direction in radians (dLat and dLon) using the node latitude (Lat0) and longitude (Lon0), the Earth's radius (R, 6378137 m), the northward velocity offset in meters (dN), and eastward offset in meters (dE) (Eqs 1 and 2). The direction the surface water travelled in radians was used to determine the latitude (Lat1) and longitude (Lon1) which represented by each hourly movement (Eqs 3 and 4), and was repeated for

24-hours until the final position of the surface water movement from each node was determined."

**R2C2** A truncated normal distribution for the process model does not make sense. The state component is the "true" log TP concentration and that is not a truncated value. A truncated normal could be used for an observation model, however.

We argue that using the truncation on the "true" log TP value is appropriate. This truncation forces the model to stay within concentration bounds that must exist in the lake. At no time step or location in the lake will "true" concentrations exceed those observed at the maximum Maumee River concentration. Additionally, the "true" value lower bound within our lake is not going to be zero or negative and a boundary condition at some logical value is needed. Additionally, most of the water entering western Lake Eire is from the Detroit River and its concentration typically ranges between 10-25 ug l$^{-1}$, amongst our 11-years of observations 0.7% of samples were under 10 ug l$^{-1}$. Thus, our 5 ug l$^{-1}$ lower bound is reasonable. Allowing the model to predict concentrations higher or lower than what we know to exist would decrease the applicability of SSMs in water quality modeling.

**R2C3** Given that year-specific SSMs were fit, it would be useful to compare the posterior distributions for the four process model parameters, $\beta$mau, $\beta$rai, $\beta$self , and $\beta$lake. At a minimum there needs to be summaries about the posteriors of these parameters, and some discussion and interpretation of the values are needed.

We will add both a visual representation of the $\beta$s, the uncertainty of the data model, the uncertainty of the process model, and a discussion of their values. The visual will be added as **Appendix C** and the discussion will be added to **Section 4.1**.

We propose the following text:
> "TP is a conservative water quality constituent. TP observations are insensitive to biogeochorus transformations of phosphorus form because these data represent both the organic and inorganic forms of phosphorus occurring in the water column. $\beta$s near 1 would then be expected in the absence of dilution. Dilution of TP would happen west to east across our spatial model window, however the depth gradient within western Lake Erie is muted. $\beta$s larger than 1 would indicate in-lake sources of TP. Every $\beta$Mau, $\beta$Ras, $\beta$Lake, and $\beta$Self fit in our models had 95% predictive intervals encompassing a value of 1. No identifiability issues identified by priors dominating the fit of coefficients were observed (Auger-Méthé et al., 2020; Appendix C).

> Fit process model or data model uncertainty were well identified (Appendix C). The apportionment of uncertainty between the process model and the data model varied from year to year. This was driven in annual variation in the data model uncertainty. We propose these annual differences were due to the combination of the number of samples collected and they relative position to the surface currents. However, the uncertainty within our models did not prevent accurate outputs estimating TP concentrations at observed and unobserved nodes."

[Figure]

Figure C1. Uninformed priors were used to fit the state space coefficients of the Maumee River ($\beta_{Mau}$, a. all values represented as red polygon), River Raisin ($\beta_{Ras}$), the western Lake Erie nodes subject to movement ($\beta_{Lake}$, b. red polygon), and those Lake Erie nodes which did not encounter sufficient water movement to associate to an "upstream" node ($\beta_{Self}$). The fitted values for every year (a. and b., all values represented as black polygon) did not significantly overlap. The log data model and process model uncertainty (c) for every year also were well identified (the uncertainty $\sigma$ was logged to aid in visually comparing prior and fitted values).

Table C1. State space models in western Lake Eire fit coefficients predicting TP concentrations from the previous time-step within the Maumee River, the River Raisin, Lake Erie, and lake locations where the current time-step is informed by the same location, $\beta_{Maumee}$, $\beta_{Raisin}$, $\beta_{Lake}$, and $\beta_{Self}$, respectively. The 95% predictive interval (PI) for each year and coefficient was examined. The same models fit annual process model and data model precision.

| Year | $\beta_{Maumee}$ 95% PI | | $\beta_{Raisin}$ 95% PI | | $\beta_{Lake}$ 95% PI | | $\beta_{Self}$ 95% PI | | Process σ 95% PI | | Data σ 95% PI | |
|------|-------|-------|-------|-------|-------|-------|-------|-------|-------|-------|-------|-------|
| 2008 | 0.989 | 1.008 | 0.985 | 1.013 | 0.986 | 0.999 | 0.963 | 1.012 | 0.263 | 0.331 | 0.02  | 0.098 |
| 2009 | 0.993 | 1.005 | 0.992 | 1.007 | 0.991 | 1     | 0.984 | 1.002 | 0.168 | 0.193 | 0.125 | 0.165 |
| 2010 | 0.994 | 1.003 | 0.993 | 1.006 | 0.988 | 0.994 | 0.992 | 1.001 | 0.13  | 0.183 | 0.326 | 0.404 |
| 2011 | 0.993 | 1.006 | 0.99  | 1.008 | 0.991 | 1.006 | 0.985 | 1.007 | 0.174 | 0.209 | 0.017 | 0.061 |
| 2012 | 0.993 | 1.007 | 0.804 | 1.186 | 0.997 | 1.008 | 0.983 | 1.005 | 0.177 | 0.218 | 0.017 | 0.062 |
| 2013 | 0.993 | 1.007 | 0.992 | 1.008 | 0.992 | 1.003 | 0.978 | 1.008 | 0.181 | 0.277 | 0.135 | 0.25  |
| 2014 | 0.992 | 1.005 | 0.99  | 1.006 | 0.991 | 1.001 | 0.991 | 1.018 | 0.173 | 0.235 | 0.145 | 0.232 |
| 2015 | 0.992 | 1.006 | 0.989 | 1.007 | 0.992 | 1.003 | 0.982 | 1.003 | 0.212 | 0.264 | 0.156 | 0.238 |
| 2016 | 0.993 | 1.005 | 0.991 | 1.005 | 0.996 | 1.004 | 0.983 | 1.001 | 0.161 | 0.2   | 0.146 | 0.214 |
| 2017 | 0.993 | 1.006 | 0.991 | 1.008 | 0.995 | 1.003 | 0.978 | 0.998 | 0.197 | 0.24  | 0.019 | 0.079 |
| 2018 | 0.992 | 1.007 | 0.99  | 1.008 | 0.983 | 0.991 | 0.966 | 0.997 | 0.221 | 0.26  | 0.018 | 0.065 |

**R2C4**  Related to item 3, a more comprehensive approach to fitting these data would be a hierarchical SSM. Model the slope coefficients (the β's) as random variables coming from a generating distribution that reflects between year (or environmental) variation; eg., $\beta_{mau,y} \sim$ Normal($\mu_{\beta mau}$; $\sigma_{\beta mau}$).

We agree that our SSM could be represented by a hierarchical form where each year is fitted together. The limitation is computation. As the reviewer notes in comment **R2C7**, the run time for each model year is very long (+/- 24 hours), so fitting all the years together would at a minimum require 18-days to run, and the outputs from such a hierarchical model would be matrices so big that exponentiating the outputs would surpass the memory available to us on our computer cluster. We will add text to **Section 4.1** suggesting a hierarchical SSM for all years as an approach readers with smaller datasets could employ.

We propose the following text:
> "This framework could be implemented with the coefficients ($\beta_{Mau}$, $\beta_{Ras}$, $\beta_{Lake}$, and $\beta_{Self}$) fit hierarchically by year, current restrictions on computer memory prevented that use here. However for smaller spatial and temporal models it could be effective."

**R2C5**  Nothing has been said about the estimates of the process model precision (Q) and the obs'n model precision (R). How do they vary between years? There are often weak identifiability problems with these kinds of linear Gaussian SSMs (see Auger-Methe et al. (2016)).

We propose added a figure as **Appendix C** with the estimates of Q and R, visually presented in our response to **R2C3**. Non-identifiability represented by overlap of the fitted values and the prior is absent across all models. The uncertainty attribution between process model and observation model differed from year to year, some years uncertainty is dominated by process error while other years it is equally attributed to process and observation. Process model precision is consistent while observation error changes year-to-year. These year-to-year variations in the observation model are potentially due to the number of samples collected or the location of the samples relative to the surface currents. We propose new text for **Section 4.1** in the above response to **R2C3**.

**R2C6**  Regarding application of the SSM to assess the effect of reducing TP loads in the Maumee River, assuming year specific SSMs were fit, then the particular SSM that was used needs to be stated. Also related to 3, it would be good to report results for each of the year specific SSMs to show (some of) the uncertainty in the assessments.

The SSMs per year were run twice, first with all the unaltered data, and second with Maumee River data halved. The values for process and observation model precision, as well as the value of every year's βs, will be added in an Appendix C as a table following the visual comparing the priors to the fitted values.

Proposed text in **Section 2.3**:
> The Maumee River impact plume was estimated by artificially reducing the Maumee River TP concentrations by 50% ($\hat{y}_{Maumee,t,y}$), each years model was then again (Eq 5-7).

Proposed text in **Section 4.1**:

> The coefficients (βs) and uncertainties (process and data model σ) varied only slightly from year to year (Table C1). The 2012 Raisin River coefficient ($β_{Ras}$) predictive interval was larger than other years because of a lack of data in that year. The proportion of uncertainty between process and data model also varied only slightly (Table C1), possibly because of the number of or spatial position of observations.

**R2C7** While it is good to have included the R code and data for fitting the SSM to the 2018 data, something needs to be said about computational time. I ran the code, but it had not completed after 14 hours, and I could not therefore verify any results.

We noted this issue in our response to **R2C4**, the model takes our system +/- 24-hours to run. We will note that within the publicly available code.

Detailed technical comments
p=page, L=line.

**R2C8** Introduction, p3, L70: Why test a hypothesis of linearity? A more general aim would be to quantify the nature of a relationship, be it linear or nonlinear.

This hypothesis is based on a conservative water quality constituent (TP) and a single source (the Maumee River). If flow is constant, the more TP imported from the Maumee River the higher concentrations should be in the lake. While this is simplistic, it is widely examined by previous research, and supporting these earlier works we found a linear relationship.

**R2C9** Methods (a) p3, L73-79. Is there overlap in the sampling on Lake Erie? For example, do two agencies collect data on the same cells?

There are instances where the same agency collects multiple samples on a single day at a single node, however no instances where differing agencies collect samples on the same day and node.

**R2C10**(b) p3, L81. By station is that just referring to the rivers, or does that include the lake? When multiple samples were collected from a \station on a single day, how much variation was there in the measure value? Such information could be used in the observation model of the SSM.

"Station" in that sentence should be replaced with "node", the multiple sampling days are rare, and the values are always very close to each other.

**R2C11**(c) pp 3-4, L84-100. It would help to create an example figure showing the geometry (or trigonometry) underlying the calculation of a particle's change in position at time t to time t + 1. For example, draw an x-y plot with points pt and pt+1 with coordinates x=longitude and y=latitude. Draw a right triangle, where the vertical and horizontal sides are parallel to the y-axis and xaxis respectively, and the hypotenuse connects pt and pt□1. So the vertical side indicates the \northerly" movement, the horizontal side indicates\easterly" movement. (For what it's worth, I _nd thinking of this as a step direction and step size process|as is done in animal movement modeling.)

A figure of an example track of surface current will be added as **Appendix D**.

[Figure]

Figure D1. Surface current data was available hourly within western Lake Erie. These 24-hour data were used to track the daily movement of water from each node.

**R2C12**(d) p3, L89. A refinement would be fit a spatially smooth velocity map rather than use values at "nodes", though I'm not sure how much practical effect that would have.

While this would be an improvement, our goal is to present a model that can be constructed easily by others by our example code.

**R2C13**(e) p4, L101. Does anything need to be handled differently at cells on the perimeter? Such cells could be transferring TP to cells outside the spatial window, presumably.

The nodes along the easter perimeter do have the additional uncertainty of missing surface water adjacencies due to their location. However, with the sources of TP on the western side of the spatial window there is little practical effect in our system. We will add text in Section 4.2 to bring this issue to the attention of readers if they have systems where perimeter issues are a larger problem.

We propose the following text:

> "Nodes along the eastern perimeter of the spatial window of our model have additional uncertainty inherent in their position. Occasionally, they will not be associated with the proper "down gradient" node because the extent removes those nodes. Within our system there is little practical effect as these nodes are far from the Maumee River and are dominated by low concentrations. This is a potential problem in other systems and may necessitate wider spatial windows to eliminate."

**R2C14**(f) p4, L109. References to SSM literature are strongly recommended; e.g., Durbin and Koopman (2012); Shumway and Stoffer (2019). Discussion of the distinction between inference about latent states and fixed parameters would be good.

The authors will insert the references suggested at their appropriate location as suggested by the reviewer. Rather than distinguishing between inference and fixed parameters here we want the reader to stay focused on our goal of defining daily TP concentrations at each node on each day irrespective of the availability of observations.

**R2C15**(g) p4, L119. Say something about the β's and what they mean. For example, regarding βself and βlake, are they likely less than 1? Is there some loss of TP between a source point k and an end point n from one day to the next? Is any sort of seasonality expected in the river values? This would make the assumption of constant βmau and βrai suspect.

We agree with the reviewer that this should be discussed further, please see our response to **R2C3**.

**R2C16**(h) p5 L126-128: The joint prior described for βmau and βrai does not exactly match what the R code indicates where the 2nd parameter in the normal is the standard deviation.

The priors described in the text will be corrected as indicated by the reviewer.

**R2C17**(i) p5, L137-138: Why is cross-validation needed to make comparisons of goodness of fit across years? Doesn't the Bayesian R2 do that? Cross-validation is more often used for model selection, which is the focus of both the referenced Vehtari et al and the Piironen and Vehtari papers.

The cross-validation via leave one-node-out compared how well the model predicted the values not available during model fitting. Estimates of $R^2$ report how well the model does while given all the available data. Cross-validation's strength is that it shows how the model output accurately gives estimates at unobserved locations, which dominate the dataset. Additionally, the cross-validation by node shows that the model does well estimating TP values irrespective of the nodes proximity to the TP river sources and that the model does well estimating TP values every year irrespective of the number or location of in-lake observations.

**R2C1**8(j) p6,L155-164: I cannot tell what is being done here. Is this necessary?

These steps describe the performance of the cross-validation as a fitted distribution. We did this so that predictive intervals about the difference between cross-validated nodes and years could be used.

**R2C19**(k) p6, L171-174: Write down an equation for deflection, $d_{n;y}$, and for the normalized estimate.

The authors will change the manuscript as suggested by the reviewer and insert the equation.

**R2C20**(l) p6, L179-185: Are 252 regressions being fit each year? Write down what effective load, $\sim_{ln;y}$ means- is it an average? Why is there a subscript n if it is measured on the Maumee? I'd be curious about identifiability issues/posterior correlations with the parameters in eq'n 11,

too| seems overly complicated. Would it make sense to take average \raw" y and take the log of that?

One regression is fit through the data. The effective load of the Maumee River (l) has a subscripts n and y because each node (n) will have its own effective load each year (y). A node close to the river would have a high effective load compared to a node far away. Similarly, a node may have a high effective load in 2008 but a low effective load in 2009 depending on direction and strength of the surface currents.

**R2C21**(m) R code: The priors for precision to the Lake, Maumee River, and River Raisin are calculated as the inverse of standard deviation; shouldn't that be the variance? Also using the data to set priors is questionable.

We have changed the way priors are set for the initial states of the Lake, Maumee, and Raisin River nodes. The first year we have available (year = 2008) is has a prior for all three node types given from values reported in Rockwell et al (2005). For all the other years the priors are defined as the mean and precision of the first 20-days of the previous year.

Rockwell, David C., et al. "The US EPA Lake Erie indicators monitoring program 1983–2002: trends in phosphorus, silica, and chlorophyll a in the central basin." *Journal of Great Lakes Research* 31 (2005): 23-34.

We propose the following text:

> "Initial conditions for the latent state $x_{n,t=1,y}$ were defined as the mean and variance of the previous year first 20 days. The first year (year = 2008) initial conditions were estimated as $N(12, 5)$ (Rockwell et al., 2005)."

**R2C22** (n) Data for R code: why does the adjacency object use have 4 values for the \next" cell, at time t + 1? Only the first is used in the code.

These columns are hold overs from our original data curation, the new adjacency matrix defined in response to **R2C1** has only one column value.

**R2C23** (o) Sensitivity analysis for the priors needs to be conducted.

The sensitivity of fitted values to the priors is now incorporated in a figure generated in response to **R2C3**.

**R2C24**3. Results (a) p7, around L185. Say something about the ranges and averages of northerly and easterly velocities (m/sec), of TP concentrations (distinguishing between Maumee River, River Raisin, and western Lake Erie), and of calculated distances moved in a single day.

In our response to **R2C11** we display an example of the water movement, we believe that further tables of northing and easting data were not aid readers in understanding our SSM approach. We will add text to report the range of TP values within each river and western Lake Erie.

**Section 3** Proposed text:

> The mean values of the observed TP concentrations within the Maumee River, River Raisin, and western Lake Erie were 170 ug l-1 (95% interval, 3.5 to 438 ug l-1), 80 ug l-1 (95%, 40 to 215 ug l-1), and 38 ug l-1 (95%, 10 to 203 ug l-1), respectively.

**R2C25**(b) p7, L195-199. In addition to the summaries in Table 1, show some plots of posterior mean values for some of the cells across the 136 day period against corresponding observations. Show an example histogram (or two) of the distribution of predicted log concentrations with the observation. Presumably these R2 are only calculated on the cells with observations. I don't understand what the cross-validated measure is doing (relates to earlier comment in methods).

Posterior mean values for a select number of nodes each year are represented in **Appendix B**. Please refer to our previous comments regarding the cross-validation in **R2C17**.

**R2C26**(c) p7, L200-205. As mentioned above need to report out results on parameter estimates for the β (each of the 4 for all 11 years) and report out the process and observation model standard deviations (1=pQ and 1=pR, based on R code).

We will generate figures depicting the values of the four βs and Q & R, for all years in an **Appendix C** as noted in our response to **R2C3**.

Minor editorial remarks

**R2C27**1. Throughout, consider using the word \cell" instead of \node" as a node is usually interpreted as a point. Node could then refer to cell center.

We prefer the term "node" but will consider using "cell" during the further editing steps in submitting the manuscript.

**R2C28**2. Section 2 Methods.
(a) pp 3-4. Consider creating a new subsection for the material in the first paragraph (L84-107) of the Model Description subsection, maybe naming it Construction of an Adjacency matrix.

We prefer the current organization as it aids the organization of results and discussion.

**R2C29**(b) p4, L109. Emphasize that 11 different SSMs will be fit.

Line 109 states "…state-space models for each year…", we will emphasize the separate years in our new table (Table C1) of Q, R, and βs.

**R2C30**(c) p4, L119. The observations are modeled with a normal distribution; they are not estimated with a normal. As mentioned previously, a truncated normal (perhaps just on the left) could be used to account for measurement limitations.

Please see our response to **R2C2**.

**R2C31**(d) p5, L139. Consider renaming Section 2.2.2 \Fitting the SSM" as \SSM Fit" could be interpreted as a result not a method.

The authors will change the manuscript as suggested by the reviewer.

**R2C32** (e) p5,L133: Perhaps move text beginning with \The model was run.." into Section 2.2.2. Could delete/move material in the sentence beginning \The efficacy of.." as it is redundant with material on L140-151.

We prefer the current organization.

**R2C33** (f) p5, L136. \efficacy" seems an odd choice, why not goodness of fit?

The authors will change the manuscript as suggested by the reviewer.

**R2C34** (g) p5,L137: The Vehtari, et al., 2017 paper does not refer to Bayesian R2. Give a mathematical definition of the R2 here: not clear to me what resolved and residual variances mean.

$R^2$ is defined on line 142. The authors will change the reference to (Gelman et al., 2019). "Resolved" is a typo, it should red "fitted".

**R2C35** (h) p5,L138 What does \utility" mean? And what value of K was used?

Please refer to our response to **R2C17**.

**R2C36** (i) p5,L145: Cross-validation is meant for CV not coefficient of variation?

The authors will change the manuscript as suggested by the reviewer.

**R2C37**(j) p5, L154: What does preferentially mean? 3

Please refer to our response to **R2C17**.

**R2C38** (k) p6, L165: Instead of Model Experimentation, wouldn't Model Usage or Application make more sense here (and elsewhere)?

We prefer the current organization.

**R2C39** (l) 6, L166: As said previously, the SSM for which year was used?

Please refer to our response to **R2C20.**

3. Section 3 Results.
**R2C40** (a) p7, L187-193. Re: the degree of missing data, I think it would be easier to follow by first saying how many space-time cells there are and then give the number with data: \For the Lake, there 252*11*136 = 376,992 cells of which 1218 had data, and for the two rivers, there were 2*11*136 = 2992 cells, of which 2258 had observations". Save the discussion of the inference for cells without data till later.

We prefer the current organization.

**R2C41**(b) p7, around L188. It would help to see an example plot that shows the spatial dist'n of cells, in a given year, that had at least one observation (see Figure 1).

**Appendix B** visually shows all the nodes which contain data through the years.

**R2C42**(c) Also a plot showing \source" cells and \end" cells would be good to show the adjacency. (Note: in the R code, use object has 4 values in the second dimension, but it appears that only the $1^{st}$ value is used as the adjacency matrix|what are the other 3 values for?)

Please refer to our response to **R2C22.**

**R2C43**(d) p7, around L188. Add a plot or two of the log(TP) concentrations. See Figure 2.

We prefer the current organization.

**R2C44** (e) p7, L195: \efficiency" here, but \efficacy" in Methods, but would model quality or goodness-of-_t be more appropriate?

The authors will change the manuscript as suggested by the reviewer.

**R2C45** (f) p7, L197: typo: Table 1 not Table 2.

The authors will change the manuscript as suggested by the reviewer.

**R2C46** (g) p7, L202: Say that Figure 2 shows 2018 and Figure B.1 in Appendix B shows 2006. The captions in those figures need to indicate that the black dots are observed values.

The authors will change the manuscript as suggested by the reviewer. See new figures below.

**R2C47** 4. p8,L222 (and p10, L294): \amending" seems an odd choice: to amend would mean to modify data in such a way that the modified data are an improvement.

We propose changing "amending" to "combining".

**R2C48** p8, L231: Was the notation k for the derived adjacency matrix used before? k was just the \source" cell from time t which feed a \sink" cell at time t + 1?

We propose deleting "(k)".

Appendix B Figures

---

## Author Comment (AC3)

Thank you for your comprehensive response to reviewers' comments. Due to the extensive nature of their comments, the reviewers will be given an opportunity to provide feedback on the proposed revisions. Please note that inclusion of additional references, suggested by the reviewers, is entirely at your own discretion. In addition to the comments provided by the reviewers, I have two small observations:

R1C1 In the Introduction, would it be possible to include a brief introduction to Bayesian state-space models and their application in freshwater ecology/ water quality modelling to set this work into a wider context for a less familiar reader, with a few key references?

We agree and propose adding text to Section 1.

> Bayesian frameworks can quantify uncertainty in the effect of nutrient load on nutrient distribution within a dynamic system such as Lake Erie. State-space models have been used in ecology to incorporate temporal and spatial autocorrelation and quantify observation error separate from the error attributable to the modelled ecological process (Auger-Méthé, 2021). State-space models are widely used in ecology to model animal populations (Buckland, 2004), movement (Royer, 2005), and fisheries stocks (Meyer and Russell, 1999). Non-stationary time-series models have been used in the Great Lakes to model water levels (Sellinger, C.E., 2008; Lamon and Stow, 2010) and to predict polychlorinated biphenyls concentration in trout (Stow, 2004). The goal of this study was to use a spatially dependent time-series state-space model approach to define concentrations at unobserved locations and quantify the impact of river nutrient delivery across western Lake Erie.

Auger-Méthé, Marie, et al. "A guide to state–space modeling of ecological time series." *Ecological Monographs* (2021).

Buckland, S. T., et al. "State-space models for the dynamics of wild animal populations." *Ecological modelling* 171.1-2 (2004): 157-175.

Royer, Francois, J-M. Fromentin, and P. Gaspar. "A state–space model to derive bluefin tuna movement and habitat from archival tags." *Oikos* 109.3 (2005): 473-484.

Lamon, E. C., and C. A. Stow. "Lake Superior water level fluctuation and climatic factors: A dynamic linear model analysis." *Journal of Great Lakes Research* 36.1 (2010): 172-178.

Meyer, Renate, and Russell B. Millar. "Bayesian stock assessment using a state–space implementation of the delay difference model." *Canadian Journal of Fisheries and Aquatic Sciences* 56.1 (1999): 37-52.

Sellinger, C.E., C.A. Stow, E C. Lamon, and S.S. Qian. 2008. Recent water level declines in the Lake Michigan-Huron system. *Environmental Science & Technology*, 42: 367-373.

Stow, C. A., E. C. Lamon, S. S. Qian, and C. A. Schrank. 2004. Will Lake Michigan lake trout meet the Great Lakes Strategy 2002 PCB reduction goal? *Environmental Science & Technology*, 38: 359-363.

R2C17 I believe that this clarification may also be useful for inclusion in the actual manuscript.

We agree and propose the following new text for **Section 2.2.2**.
> "The cross-validation via leave one-node-out was used to evaluate model predictions where observations are not available. This estimate of model performance is needed as our dataset has far fewer observations than the product of nodes and timepoints. Estimates of $R^2$ report how well the model does while given all the available data. Additionally, the cross-validation estimates aggregated by unobserved node (space) or by year (time), defines how well the model estimates TP values irrespective of the node's proximity to the TP river sources or the number and location of annual in-lake observations."

---

## Referee Report (RR1)

**Review of the Revision of:**

**Spatially Referenced Bayesian State-Space Model of Total Phosphorus in western Lake Erie**

November 11, 2021

Referee: Ken Newman, Biomathematics & Statistics Scotland, and School of Mathematics, University of Edinburgh

**1 Overall comments**

The manuscript remains generally well organized and written. The authors have responded fairly thoroughly to the comments made in the first review, and I appreciate them checking out what happens with hourly step sizes to construct the adjacency matrix. Several substantive concerns remain, however.

**1.1 All $\beta$'s in process model near 1 and consequence**

That the posterior distributions for all four $\beta$'s in the process model are so concentrated near 1 (across all years) seems remarkable. That coincidence aside, assuming that the latent states are the logarithms of the true TP value, the effects on the expected TP transfer from a "source" node (denoted $k$) to a corresponding "sink" or recipient node (denoted $n$) implies an increase in TP at the sink node on the raw scale. Letting $z_{n,t,y}$ be the raw scale TP value, thus $x_{n,t,y,} = \ln(z_{n,t,y})$, then the expected TP value in the sink node

$$E[z_{n,t,y}|z_{k,t,y}] = \exp\left(\beta + \ln(z_{k,t,y}) + \tau^2/2\right)$$

Substituting $\beta=1$ and $\tau^2=0.2^2$ (a rough average from Table C1):

$$E[z_{n,t,y}|z_{k,t,y}] = \exp\left(1 + \ln(z_{k,t,y}) + 0.2^2/2\right) = 2.77 z_{k,t,y}$$

**1.2 Posterior summaries**

Another concern is with regard to the posteriors for the process and obs'n standard deviations. In Figure C1, the ranges of the joint priors for the pairs $(\beta_{rai}, \beta_{mau})$, $(\beta_{self}, \beta_{lake})$, and $(\ln(\sigma_y), \ln(\tau_x))$ (my added subscripts) are denoted by the red convex hulls. The black polygons denote posterior fitted values (for all 11 years).

- Are fitted values posterior means?

- I don't understand the sentence in the figure caption about the fitted values for each year not overlapping. Based on Table C1 there is considerable similarity in the $\beta$'s: they are all very close to 1.

- With Figure C1.c it is disconcerting to show posterior means outside the support of the prior; that should not be so.

- Based on the `R` code the priors for the precision for the process and obs'n models ($1/\tau^2$ and $1/\sigma^2$) are Gamma(0.001, 0.001). The ranges for $\tau$ and $\sigma$ are the positive real numbers (which in practice, see below, can be 0 and $\infty$), and the subsequent range for $\ln(\tau)$ and $\ln(\sigma)$ is much larger than Figure C1.c indicates (roughly -6 to 11).

```
set.seed(301)
n <- 10000
Q     <- rgamma(n,0.001,0.001)  # state
sd.Q <- 1/sqrt(Q)
ok.Q <- sd.Q != 0 & !is.infinite(sd.Q)
summary(log(sd.Q[ok.Q]))
**Min. 1st Qu.  Median     Mean 3rd Qu.     Max.**
**-3.705  66.940 148.888 159.015 243.251 368.766**
```

**1.3 Identifiability of $\sigma^2$ and $\tau^2$**

Potential identifiability issues for $\tau$ and $\sigma$ have not been addressed. What needs to be examined is the correlation between $\tau$ and $\sigma$. Scatterplots of sampled pairs from the posterior distribution, at a minimum, need to be examined (and shown). Figure 1 shows how the lower and upper bounds of the PIs relate, suggesting a negative association.

[Figure]

Figure 1: Scatterplots of lower and upper bounds of 95% PIs for $\tau$ and $\sigma$. p.lo and p.up are lower and upper endpoints for the process standard deviation; similarly, o.lo and o.up for observation.

**1.4 Bayesian $R^2$ calculations**

My understanding of the Bayesian $R^2$ calculations based on Gelman et al (2019, eq'n 3) is that the calculated value of $var_{res}$ would be the estimated value of the observation variance terms, $(\sigma^2)^s$ for sample $s$ from the posterior. What is written in the text seems to match the Gelman et al eq'n 2 definition for *non*-Bayesian $R^2$. How much difference that makes needs to be examined—the fact that the majority of the average Bayesian $R^2$ are greater than 0.98 or so seems remarkable.

**1.5 Cross-validation calculation and subsequent modelling of scores**

The description of the K-Fold cross-validation score in eq'n 8 is unclear. The summation index is $d$ but the last value of the index is also $d$, the term $d$ does not appear in the values being summed, and the conditioning notation $\tilde{y}_{n,t,y}$ is is ambiguous. I am assuming that the eq'n is patterned after equation 3 in Piironen and Vehtari (2017) and conditioning would be on the set excluding the $y_{n,t,y}$ values; something like they used $D_{-y_{n,t,y}}$ might be more understandable.

The cross-validation scores are then regressed on $\mu_{o,n}$, $\mu_n$'s (eq'n 9), or $\mu_{o,y}$ and $\mu_y$ (eq'n 10), where these regressors are then given priors. The terms on the left-hand sides of eq'ns 9 and 10 need subscripting. More critically, are the regressors pure random effects and what are called priors actually the probability distributions for these random effects? I tried to find where in Kruschke (2014; that should be 2015 I believe) such a procedure was used—chapter or page numbers need to indicated.

**2 Detailed technical comments**

p=page, L=line.

1. Methods

    (a) p3, L93: the 2.2 heading "Model description" still seems misleading, as the real model is the SSM. Subsection 2.2 is about creating the adjacency matrix, which is *not* a model, per se.

    (b) Constraints on State process values: Need to state that $x_{n,t,y}$ is the logarithm of TP: this is also important as the authors' reply about TP not being negative applies to the raw scale values, not the logarithm. While I can see the argument for constraining the range of $x_{n,t,y}$ based on expert opinion, I don't think that the observations in a given year should be used to determine those constraints: the prior needs to be independent of the data, nor does it make sense to me that measuring instrument limits constrain the true value. Why not simply say that our prior opinion is that log of TP lies between $a$ and $b$ and use the same values for each year?

    I am possibly misinterpreting the scale for the obs'ns but based on the R code: `concs = log(concs*10`$^3$`)`, summaries of raw and logged TP values for the Maumee River, Raisin River, and the lake are the following:

    |  | Min. | 1st Qu. | Median | Mean | 3rd Qu. | Max. | NA's |
    |---|---|---|---|---|---|---|---|
    | Maumee R | 77.33 | 173.38 | 218.07 | 261.20 | 334.25 | 804.80 | 21.00 |
    | Raisin R | 28.00 | 65.95 | 85.15 | 107.12 | 109.35 | 458.20 | 65.00 |
    | Lake | 5.50 | 20.60 | 33.20 | 51.05 | 63.81 | 275.07 | 91789.00 |
    | log Maumee R | 4.35 | 5.16 | 5.38 | 5.44 | 5.81 | 6.69 | 21.00 |
    | log Raisin R | 3.33 | 4.19 | 4.44 | 4.51 | 4.69 | 6.13 | 65.00 |
    | log Lake | 1.70 | 3.03 | 3.50 | 3.62 | 4.16 | 5.62 | 91789.00 |

    The authors' reply that 0.7% were less than 10 $ug/l$ is presumably referring to the lake samples; I found one out of 191 of the lake values less than 10. Note that the above means differ from those reported in the Results (lines 214-215), but maybe I've made mistakes.

    (c) p4, L122-135: I find the explanation of the SSM awkward. Need to state somewhere that the latent states are log transformed (only by checking the R code did that become clear). Consider: "The SSM consists of two models, one for the data ($y$) called the observation model and one for latent states ($x$) called the process model." The observations were modelled as follows.

    $$y_{n,t,y} \sim \text{Normal}\left(x_{n,t,y}, \sigma^2\right) \tag{1}$$

    where $y_{n,t,y}$ is the natural logarithm of the measured TP concentration at node $n$ on day $t$ of year $y$, $x_{n,t,y}$ is latent (unobserved) true log TP concentration, and $\sigma^2$ is the observation standard

deviation. The process model is first order Markov, only depending on the value of the node at time $t-1$ which transported to TP to node $n$ at time $t$. That source node is denoted $k$ and for nodes in the river, $k=n$, and for nodes in the lake, $k$ is determined from the time $t$ adjacency matrix.

$$x_{n,t,y} \sim \text{Truncated Normal}\left(f(x_{k,t-1,y}), \tau^2\right) \quad I(a \leq x_{n,t,y} \leq b), \tag{2}$$

where

$$f(x_{k,t-1,y}) = \begin{cases} \beta_{mau} * x_{k,t-1,y} & \text{if } n=\text{Maumee River node} \\ \beta_{rai} * x_{k,t-1,y} & \text{if } n=\text{River Raisin node} \\ \beta_{self} * x_{k,t-1,y} & \text{if } n=\text{same lake node} \\ \beta_{lake} * x_{k,t-1,y} & \text{if } n=\text{a different lake node} \end{cases} \tag{3}$$

The process standard deviation is $\tau$ and the values for $x_{n,t,y}$ are restricted to $[a,b]$

Also please write the eq'ns for the priors for the $\beta$'s to ease comprehension for the reader. It's not clear to me what the sentence at lines 136-137 is saying. Only by examining the R code could I tell that:

$$\beta_{self}, \beta_{lake} \overset{iid}{\sim} \text{Normal}\,(0, 10,000)$$
$$\beta^* \sim \text{Normal}\,(0, 10,000)$$
$$\tau_{mau}, \tau_{rai} \overset{iid}{\sim} \text{Gamma}\,(0.001, 0.001)$$
$$\beta_{mau} \sim \text{Normal}\,\left(\beta^*, \tau_{mau}^{-2}\right)$$
$$\beta_{rai} \sim \text{Normal}\,\left(\beta^*, \tau_{rai}^{-2}\right)$$

Also write the priors for $1/\tau^2$ and $1/\sigma^2$.

2. Results. Major comments about the results were given in Section 1. One other point is the regression of mean TP concentration on effective Spring TP load: doesn't distance of the lake nodes from the Maumee River have an effect on the relationship? Distance is not included in the regression.

3. Discussion. Not clear what in Auger-Methe, et al (2021) is being referred to regarding identifiability and visual determination of priors dominating. Is what is meant that the posterior and prior will not differ much for parameters that are unidentiable or weakly identifiable?

**Minor editorial remarks**

1. Abstract

   (a) p1, L22: "estimated that, in the absence of the Maumee River load, lake concentrations..."

2. Introduction

   (a) p2, L46: "affects"

   (b) p2, L52: here refer to soluble reactive phosphorous (SRP), but later (starting with p3, L80) refer to TP, without defining what TP means. I'm no expert on water chemistry, but my understanding is that TP, total phosphorous, includes SRP. TP needs to be defined, and if SRP is not referred to again, perhaps do not add the abbreviation.

   (c) p2, L60: perhaps "Bayesian inference" instead of "Bayesian frameworks". This paragraph is more about SSMs than about Bayesian inference, and it might be better to make a statement about SSMs first (the topic sentence), e.g., use the 2nd sentence without the adjective Bayesian: "State-space models (SSMs) have been used...". Then discuss the application areas and then add a sentence or two about Bayesian SSMs.

(d) p3, L68: "While spatial models"

(e) p3, L71: "incorporate concentration data"

(f) p3, L80: define TP

3. Methods

   (a) p4, L100-110. Perhaps: "Hourly northward and eastward transport ... was expressed in radians:

$$dLat_t = \frac{dN_t}{R}$$

$$dLon_t = \frac{dE_t}{R \cos\left(\pi \frac{Lat_t}{180}\right)}$$

   Then add "The latitude and longitude at time $t + 1$, given the latitude and longitude at time $t$ and the above derivatives, was calculated as follows:

$$Lat_{t+1} = Lat_t + dLat_t * \left(\frac{180}{\pi}\right)$$

$$Lon_{t+1} = Lon_t + dLon_t * \left(\frac{180}{\pi}\right)$$

   I don't think Eqs 1-4 ever get referred to and they do not need to be numbered. I found the use of 0 and 1 confusing as this procedure is carried out at every time step.

   (b) p5, L145. The material beginning "The model was run" belongs in a section labelled Fitting the SSM, not in the SSM model description section. It might be more appropriate to name Section 2.2.2 SSM fitting and diagnostics.

---

## Author Response (AR2)

E1C1

Thank you for your revised manuscript, which has now been reviewed by two reviewers. The first reviewer still has concerns regarding the mathematical implementation of the model. A particular concern relates to the estimated slope coefficients relating transfers of the logarithm of TP from a source node to a recipient node being near 1, with the resulting implications for transfers on the raw (un-logged) scale increasing by about 2.8. The reviewer also has concerns regarding the calculations of the Bayesian R-sq values and why they are so high and would like to see a clearer explanation of the K-fold cross-validation calculation. The reviewer likes the idea of using a state-space model for this application but has concerns regarding the implementation of the approach. I agree with both reviewers that the application of the SSM approach to the understanding of total phosphorus pollution in a large lake system is novel and worth pursuing. Could you please therefore take careful account of the constructive criticism offered in the referee 1 report and consider revising your manuscript to address the raised concerns?

Dr. Glendell,

Thank you for the opportunity to respond to the peer reviewed comments. We have addressed each in the attached point-by-point response document and noted changes in the attached "tracked changes" manuscript copy with each reviewer comment (R1C1; reviewer 1 comment 1) noted.

Reviewer 1's particular concern stemming from the coefficients near 1 resulting in a doubling of values is not taking place in the model. We show unequivocally that the values do not double, moreover the code and data we provide at https://doi.org/10.5281/zenodo.5570508 can be run by anyone and will show no such doubling. The Bayesian R-sq values are so high because we use a daily time-step and the daily data available in the two river datasets create inflated R-sq values as the model passes through these observations with a very restricted predictive interval (illustrated in Figure 2a and 2d). This underscores the importance of using a leave-one-out K-fold cross validation. The nomenclature of the K-fold cross validation has been updated and the reference updated to provide the page number. Our explanation of the K-fold cross-validation calculation shows how we are removing a node with observations, predicting those observations, and ensuring that predictions are preforming consistently for each year and irrespective of which node was removed. Additionally, a discussion of Reviewer 2's concerns with dilution and settling were updated in the text.

As authors, we are pleased to submit this manuscript and feel the extensive feedback from the previous rounds of review have made a complete story we are eager to see finished. Please contact me with any subsequent questions. Thank you for your consideration.

Reviewer #1 (For easy of review, the comments from Reviewer #1 are pasted as images rather than coping the text off the submitted PDF)

Reviewer Comments – Black

Author Response – Green

Altered text – Blue

Page numbers referred to the "Tracked Changes" version of manuscript.

R1C1

The manuscript remains generally well organized and written. The authors have responded fairly thoroughly to the comments made in the first review, and I appreciate them checking out what happens with hourly step sizes to construct the adjacency matrix. Several substantive concerns remain, however.

That the posterior distributions for all four $\beta$'s in the process model are so concentrated near 1 (across all years) seems remarkable. That coincidence aside, assuming that the latent states are the logarithms of the true TP value, the effects on the expected TP transfer from a "source" node (denoted $k$) to a corresponding "sink" or recipient node (denoted $n$) implies an increase in TP at the sink node on the raw scale. Letting $z_{n,t,y}$ be the raw scale TP value, thus $x_{n,t,y,} = \ln(z_{n,t,y})$, then the expected TP value in the sink node

$$E[z_{n,t,y}|z_{k,t,y}] = \exp\left(\beta + \ln(z_{k,t,y}) + \tau^2/2\right)$$

Substituting $\beta=1$ and $\tau^2=0.2^2$ (a rough average from Table C1):

$$E[z_{n,t,y}|z_{k,t,y}] = \exp\left(1 + \ln(z_{k,t,y}) + 0.2^2/2\right) = 2.77 z_{k,t,y}$$

Values are logged prior to model fitting and remain in log space until the samples of the fitted posteriors are exponentiated.

E.g., a value of 100 ug/l total phosphorus

$y_{t=1} = \log(100) = 4.60517$

Sampling State Space Model: $4.60517 * \mathbf{1} + \text{data/process error} \approx 4.62$

$\hat{y}_{t=2} = \exp(4.62) = 101.5$ ug/l total phosphorus

There is no doubling of concentrations between nodes. Please refer to R1C8.

**R1C2**

Another concern is with regard to the posteriors for the process and obs'n standard deviations. In Figure C1, the ranges of the joint priors for the pairs $(\beta_{rai}, \beta_{mau})$, $(\beta_{self}, \beta_{lake})$, and $(\ln(\sigma_y), \ln(\tau_x))$ (my added subscripts) are denoted by the red convex hulls. The black polygons denote posterior fitted values (for all 11 years).

- Are fitted values posterior means?

- I don't understand the sentence in the figure caption about the fitted values for each year not over-lapping. Based on Table C1 there is considerable similarity in the $\beta$'s: they are all very close to 1.

- With Figure C1.c it is disconcerting to show posterior means outside the support of the prior; that should not be so.

- Based on the R code the priors for the precision for the process and obs'n models ($1/\tau^2$ and $1/\sigma^2$) are Gamma(0.001, 0.001). The ranges for $\tau$ and $\sigma$ are the positive real numbers (which in practice, see below, can be 0 and $\infty$), and the subsequent range for $\ln(\tau)$ and $\ln(\sigma)$ is much larger than Figure C1.c indicates (roughly -6 to 11).

- Fitted values are not represented as means, these black polygons are the extent of all sampled posteriors for all years.
- This sentence has been revised

Lines 446 to 447

The fitted values for every year (a. and b., all values represented as black polygon) do not appear to be overly influenced by the uninformed priors.

- The posterior values of Figure C1(c) are not disconcerting as the priors are purposefully broad and uninformed.
- The values in the figures are where generated via JAGS sampling of the defined uninformed priors, they do not represent the breath of theoretical values.

**R1C3**

Potential identifiability issues for $\tau$ and $\sigma$ have not been addressed. What needs to be examined is the correlation between $\tau$ and $\sigma$. Scatterplots of sampled pairs from the posterior distribution, at a minimum, need to be examined (and shown). Figure 1 shows how the lower and upper bounds of the PIs relate, suggesting a negative association.

[Figure]

Figure 1: Scatterplots of lower and upper bounds of 95% PIs for $\tau$ and $\sigma$. p.lo and p.up are lower and upper endpoints for the process standard deviation; similarly, o.lo and o.up for observation.

The lack of σ and τ correlation is adequately displayed in Figure C1(c), which are the scatter plots that the reviewer specifies.

**R1C4**

My understanding of the Bayesian $R^2$ calculations based on Gelman et al (2019, eq'n 3) is that the calculated value of $var_{res}$ would be the estimated value of the observation variance terms, $(\sigma^2)^s$ for sample $s$ from the posterior. What is written in the text seems to match the Gelman et al eq'n 2 definition for $non$-Bayesian $R^2$. How much difference that makes needs to be examined—the fact that the majority of the average Bayesian $R^2$ are greater than 0.98 or so seems remarkable.

The estimates of Bayesian R2 have been recalculated via Gelman et al (2019) Eq 3, and values updated in Table 1. This approach only made our Bayesian R2 values closer to 1. The reason this model metric looks so "good" is discussed in the additional text below.

Bayesian R2 defined as the fitted variance (varfit) divided by the sum of varfit and the residual variance (varres) was calculated for each model year. Model varfit was the variance of the modelled predictive mean, while varres is estimated by squared standard deviation of the errors (Gelman et al., 2019).

Lines 263 to 274

The models consistently generated plausible posterior samples for mean TP concentration as each 95% CI of annual posterior predictive p-values included 0.5 and annual Bayesian R2 95% CI values ranged from 0.84 to 0.99 (Table 1). Annual posterior predictive p-values indicate that our model framework is preforming well predicting water quality within large water bodies even with sparse observations within the data. While our high Bayesian R2 values appear to support the use of our model, it is likely that they represent an inappropriate model metric. The state-space framework forces the model to pass through the observed data and the daily observation in the river datasets are likely driving Bayesian R2 values higher with their constrained predictive intervals. Because of these elevated Bayesian R2 estimates, the k-fold CV are important checks on the applicability of this state-space approach. The k-fold CV results generated by removing all the observations of a randomly selected lake node with at least 10 observations showed that model predictions were equally accurate across years and by node. Predicting equally well across the nodes and within any year provides strong support for this framework as being a useful application of Bayesian methods in water quality modelling.

**R1C5**

The description of the K-Fold cross-validation score in eq'n 8 is unclear. The summation index is $d$ but the last value of the index is also $d$, the term $d$ does not appear in the values being summed, and the conditioning notation $\bar{y}_{n,t,y}$ is is ambiguous. I am assuming that the eq'n is patterned after equation 3 in Piironen and Vehtari (2017) and conditioning would be on the set excluding the $y_{n,t,y}$ values; something like they used $D_{-y_{n,t,y}}$ might be more understandable.

The cross-validation scores are then regressed on $\mu_{o,n}$, $\mu_n$'s (eq'n 9), or $\mu_{o,y}$ and $\mu_y$ (eq'n 10), where these regressors are then given priors. The terms on the left-hand sides of eq'ns 9 and 10 need subscripting. More critically, are the regressors pure random effects and what are called priors actually the probability distributions for these random effects? I tried to find where in Kruschke (2014; that should be 2015 I believe) such a procedure was used—chapter or page numbers need to indicated.

The subscripts in Eqn 8 have been updated to reflect that the log predictive density from posterior samples come from the $k$th node which were observations were removed and subsequently fit. This is important because was we discuss in R1C4 the other measures of model performance do not give a complete accounting of how the model does within the lake where data is scarce. "d" represents the omitted observations. As per the sentences preceding Eqn 8. We are choosing to keep our $y_{k,t,y}$ nomenclature to remain consistent with the previously described data and process model.

The K-Fold CV's have been given subscripts defining the cross validation done by node "*n*" or year "*y*". The variables fit in Eq 9 and 10 are the between-group and among-group variance as described in "Doing Bayesian Data Analysis" page 560, the reference has been updated to reflect the correct year and page number.

**R1C6**

p3, L93: the 2.2 heading "Model description" still seems misleading, as the real model is the SSM. Subsection 2.2 is about creating the adjacency matrix, which is *not* a model, per se.

We do not feel the current headings are misleading.

**R1C7**

Constraints on State process values: Need to state that $x_{n,t,y}$ is the logarithm of TP: this is also important as the authors' reply about TP not being negative applies to the raw scale values, not the logarithm. While I can see the argument for constraining the range of $x_{n,t,y}$ based on expert opinion, I don't think that the observations in a given year should be used to determine those constraints: the prior needs to be independent of the data, nor does it make sense to me that measuring instrument limits constrain the true value. Why not simply say that our prior opinion is that log of TP lies between $a$ and $b$ and use the same values for each year?

I am possibly misinterpreting the scale for the obs'ns but based on the R code: `concs = log(concs*10³)`, summaries of raw and logged TP values for the Maumee River, Raisin River, and the lake are the following:

|  | Min. | 1st Qu. | Median | Mean | 3rd Qu. | Max. | NA's |
|---|---|---|---|---|---|---|---|
| Maumee R | 77.33 | 173.38 | 218.07 | 261.20 | 334.25 | 804.80 | 21.00 |
| Raisin R | 28.00 | 65.95 | 85.15 | 107.12 | 109.35 | 458.20 | 65.00 |
| Lake | 5.50 | 20.60 | 33.20 | 51.05 | 63.81 | 275.07 | 91789.00 |
| log Maumee R | 4.35 | 5.16 | 5.38 | 5.44 | 5.81 | 6.69 | 21.00 |
| log Raisin R | 3.33 | 4.19 | 4.44 | 4.51 | 4.69 | 6.13 | 65.00 |
| log Lake | 1.70 | 3.03 | 3.50 | 3.62 | 4.16 | 5.62 | 91789.00 |

The authors' reply that 0.7% were less than 10 $ug/l$ is presumably referring to the lake samples; I found one out of 191 of the lake values less than 10. Note that the above means differ from those reported in the Results (lines 214-215), but maybe I've made mistakes.

The values within the Lake and within the Maumee River are too highly variable for a constant constraint between two values to be useful. Truncating the values of the latent state does not set a prior based on the data, rather it constrains the latent state within each year to values that represent reality based on the source dataset. The effectiveness of our approach is evident in Figure 4 where the culmination of our modeling scheme generates the a linear expression of Maumee River TP spring load to western Lake Erie TP concentration.

The values the reviewer generated the above table were only from 2018 which we provided in the online repository as an example. The values in the manuscript are representative of all our available data.

**R1C8**

p4, L122-135: I find the explanation of the SSM awkward. Need to state somewhere that the latent states are log transformed (only by checking the R code did that become clear). Consider: "The SSM consists of two models, one for the data ($y$) called the observation model and one for latent states ($x$) called the process model." The observations were modelled as follows.

$$y_{n,t,y} \sim \text{Normal}\left(x_{n,t,y}, \sigma^2\right) \tag{1}$$

where $y_{n,t,y}$ is the natural logarithm of the measured TP concentration at node $n$ on day $t$ of year $y$, $x_{n,t,y}$ is latent (unobserved) true log TP concentration, and $\sigma^2$ is the observation standard deviation. The process model is first order Markov, only depending on the value of the node at time $t-1$ which transported to TP to node $n$ at time $t$. That source node is denoted $k$ and for nodes in the river, $k=n$, and for nodes in the lake, $k$ is determined from the time $t$ adjacency matrix.

$$x_{n,t,y} \sim \text{Truncated Normal}\left(f(x_{k,t-1,y}), \tau^2\right) \quad I(a \leq x_{n,t,y} \leq b), \tag{2}$$

where

$$f(x_{k,t-1,y}) = \begin{cases} \beta_{mau} * x_{k,t-1,y} & \text{if } n=\text{Maumee River node} \\ \beta_{rai} * x_{k,t-1,y} & \text{if } n=\text{River Raisin node} \\ \beta_{self} * x_{k,t-1,y} & \text{if } n=\text{same lake node} \\ \beta_{lake} * x_{k,t-1,y} & \text{if } n=\text{a different lake node} \end{cases} \tag{3}$$

The process standard deviation is $\tau$ and the values for $x_{n,t,y}$ are restricted to $[a,b]$

Also please write the eq'ns for the priors for the $\beta$'s to ease comprehension for the reader. It's not clear to me what the sentence at lines 136-137 is saying. Only by examining the R code could I tell that:

$$\beta_{self}, \beta_{lake} \overset{iid}{\sim} \text{Normal}(0, 10,000)$$
$$\beta^* \sim \text{Normal}(0, 10,000)$$
$$\tau_{mau}, \tau_{rai} \overset{iid}{\sim} \text{Gamma}(0.001, 0.001)$$
$$\beta_{mau} \sim \text{Normal}\left(\beta^*, \tau_{mau}^{-2}\right)$$
$$\beta_{rai} \sim \text{Normal}\left(\beta^*, \tau_{rai}^{-2}\right)$$

Also write the priors for $1/\tau^2$ and $1/\sigma^2$.

The text immediately after the model definitions in our revised text noted the logged concentrations. To again emphasize this, we have added text to describe how the data were logged prior to fitting the model and that output MCMC samples were not exponentiated until plotting. Thank you for the suggested text however we intend to keep the current revised version, The priors for σ and τ were uniformed gamma distributions which are now in the text.

Lines 135 to 136

The data were logged prior to fitting the model and MCMC samples remained in log space until exponentiated for plotting.

Lines 147 to 150

The latent state ($x_{n,t,y}$; Eq 6) is sampled from a normal distribution of a predicted latent state ($xp_{n,t,y}$, Eq 7) and standard deviation $\tau$. $x_{n,t,y}$ was truncated by the detection limit of TP laboratory analysis (5 µg l$^{-1}$, $a$, Eq 6) and the maximum value observed in each year ($y$) within the Maumee River ($b$, Eq 6), $xp_{n,t,y}$ was defined depending on the node $n$ as being a river or lake node (Eq 7), and $\sigma$ and $\tau$ were fit with uniformed $gamma$(0.001,0.001) priors.

R1C9

Results. Major comments about the results were given in Section 1. One other point is the regression of mean TP concentration on effective Spring TP load: doesn't distance of the lake nodes from the Maumee River have an effect on the relationship? Distance is not included in the regression.

Distance is wrapped up in the deflection of the predictive intervals through time because it isn't just the distance but also whether the water mass from the Maumee physically moves toward a node. There are several days in which even the closer nodes are bypassed because the currents take Maumee River water in a different direction. The dual complications of distance and movement have complicated previous attempts at defining a single relationship between Maumee load and observed in-lake concentrations, which we overcome here.

R1C10

Discussion. Not clear what in Auger-Methe, et al (2021) is being referred to regarding identifiability and visual determination of priors dominating. Is what is meant that the posterior and prior will not differ much for parameters that are unidentiable or weakly identifiable?

Yes, as we discussed in our prior revisions, identifiability as defined by the priors dominating the parameter fit was not observed.

R1C11

p1, L22: "estimated that, in the absence of the Maumee River load, lake concentrations..."

Change made

R1C12

p2, L46: "affects"

Change made

R1C13

p2, L52: here refer to soluble reactive phosphorous (SRP), but later (starting with p3, L80) refer to TP, without defining what TP means. I'm no expert on water chemistry, but my understanding is that TP, total phosphorous, includes SRP. TP needs to be defined, and if SRP is not referred to again, perhaps do not add the abbreviation.

We have made these changes.

Lines 53 to 54

Spring Maumee River soluble reactive phosphorus export correlates with western Lake Erie HABs extent; this pattern has been observed since the soluble reactive phosphorus loads started to increase in the 1990s (Ho and Michalak, 2017; Michalak et al., 2013; Stow et al., 2015).

Lines 82 to 84

Here, we quantified how well our model fits the data and generated predictions of total phosphorus (TP) concentrations across western Lake Erie. TP includes the dissolved and particulate forms of phosphorus.

R1C14

p2, L60: perhaps "Bayesian inference" instead of "Bayesian frameworks". This paragraph is more about SSMs than about Bayesian inference, and it might be better to make a statement about SSMs first (the topic sentence), e.g., use the 2nd sentence without the adjective Bayesian: "State-space models (SSMs) have been used...". Then discuss the application areas and then add a sentence or two about Bayesian SSMs.

We have made these changes.

Lines 63 to 70

Bayesian state-space models have been used in ecology to incorporate temporal and spatial autocorrelation and quantify observation error separate from the error attributable to the modelled ecological process (Auger-Méthé et al., 2021; Durbin and Koopman, 2012; Shumway and Stoffer, 2019). State-space models are widely used in ecology to model animal populations (Buckland et al., 2004), movement (Royer et al., 2005), and fisheries stocks (Meyer and Millar, 1999). Bayesian inference can quantify uncertainty in the effect of nutrient load on nutrient distribution within a dynamic system such as Lake Erie. Non-stationary time-series models have been used in the Great Lakes to model water levels (Lamon and Stow, 2010; Sellinger et al., 2008) and to predict polychlorinated biphenyls concentration in trout (Stow et al., 2004).

**R1C15**

p3, L68: "While spatial models"

Please refer to R2C4.

**R1C16**

p3, L71: "incorporate concentration data"

Change made

**R1C17**

p3, L80: define TP

Please refer to R1C13.

**R1C18**

p4, L100-110. Perhaps: "Hourly northward and eastward transport ... was expressed in radians:

$$dLat_t = \frac{dN_t}{R}$$

$$dLon_t = \frac{dE_t}{R \cos\left(\pi \frac{Lat_t}{180}\right)}$$

Then add "The latitude and longitude at time $t + 1$, given the latitude and longitude at time $t$ and the above derivatives, was calculated as follows:

$$Lat_{t+1} = Lat_t + dLat_t * \left(\frac{180}{\pi}\right)$$

$$Lon_{t+1} = Lon_t + dLon_t * \left(\frac{180}{\pi}\right)$$

I don't think Eqs 1-4 ever get referred to and they do not need to be numbered. I found the use of 0 and 1 confusing as this procedure is carried out at every time step.

We have changed Eq 3 and 4 to use the "t" notation, "t" is already defined at the beginning of that paragraph, so we did not need to add the sentence suggested. Eq 1 and 2 are referred to in the preceding paragraph, so their numbered designations remain.

R1C19

p5, L145. The material beginning "The model was run" belongs in a section labelled Fitting the SSM, not in the SSM model description section. It might be more appropriate to name Section 2.2.2 SSM fitting and diagnostics.

Our opinion that the sentence should remain in Section 2.2.1. because it precedes the description of iterations, thin, and chain numbers, all consistent with what reader would expect following the model parameterization description at the beginning of this paragraph. We have added "Diagnostics" to the Section 2.2.2. heading.

Reviewer #2

R2C1 This manuscript presents an interesting application of state-space models for predicting the distribution of TP across a large body of water. The presentation is clear, and the resulting model provides a useful contribution to the literature. I have only minor comments. Spatial resolution: The selection of the 2 km x 2 km grid seems to derive from the spatial scale of currents. Is this a correct assumption? Have other spatial resolutions been considered for discretizing the distribution of TP concentrations? For example, a coarser resolution might be necessary to feasibly model a larger body of water, whereas a finer resolution might enable one to perceive smaller scale features. Or, is the modeling resolution strictly determined by the spatial scale of the current database?

For our application, yes the 2km x 2km grid was chosen to match the surface current dataset. While we did not experiment with other discrete grid distances, any applicable configuration will work. Defining a reasonable grid distance could be based on the spatial distribution of the available data and a willingness to extrapolate or average surface current direction and magnitude. Similarly, our temporal time-step was daily, but this could also be applied to monthly data in larger data sparse systems or hourly data in smaller data rich applications. This spatial and temporal flexibility or using state space frameworks gives users the capacity to tune the computational runtime and resolution of models to fit the hypothesis tested.

R2C2 Dilution from Maumee to Erie: I would have guessed that there would be some dilution of Maumee River flow as it enters Lake Erie, with an associated decrease in TP concentration. However, the coefficient Beta_mau is 1, indicating no change in concentration. Is this because the Maumee River flow accounts for the majority of the near shore water volume?

The Maumee coefficient (like the Raisin and two lake coefficients) was close to 1 because on a daily time-step the TP concentrations do not widely vary. E.g., the concentration today is similar to the concentration yesterday. The uncertainty in the process and data models allows the model predictions to trend toward the observations where available and be constrained where previous time-steps passed through observations. Were these coefficients to exceed 1 this would be evidence of other inputs of P or less that 1 would indicate some internal loss such as settling. TP is conservative to processes within the water column because it accounts for the dissolved and particulate P, if our model was applied only to dissolved P which is subject to strong assimilation pressure by phytoplankton the model coefficients would be negative, this could be very useful to others, so we maintain the coefficient use here.

R2C3 Line 31: "Excessive nutrient export primarily from agricultural watersheds…" The phrasing of this sentence seems to imply that eutrophication can mostly be attributed to agricultural nutrient loads, which may be true for certain receiving waters, but is not true for all water bodies.

We have altered that sentence to frame it as an example of the point and non-point sources introduced in the previous sentence.

Lines 30 to 31

"Observed concentrations are driven by both point and non-point sources. E.g., Wastewater effluent and excessive nutrient export primarily from agricultural watersheds may lead to eutrophication, harmful algae blooms (HABs), and threatens drinking water contamination (Brooks et al., 2016; Mellios et al., 2020; Schneider and Bláha, 2020)."

R2C4 Line 68: "Spatial" does not need to be capitalized.

"Spatial" in this sentence does not need to be capitalized.

Lines 74 to 75

"While spatial models have been used in the Great Lakes for predicting HABs biomass, HABs extent, and nutrient transport (Fang et al., 2019; Schwab et al., 2009), we proposed a Bayesian framework for similar spatial data."

R2C5 Line 77: Might want to break this section into at least two sentences.

We have broken this sentence into three to aid the reader interpreting what model we built and how we used the model to test a hypothesis.

Lines 82 to 87

"Together our contribution will fit values in the absence of observations and allow experimentation in archival data previously not possible. Here, we quantified how well our model fits the data and generated predictions of TP concentrations across western Lake Erie. Additionally, we experimentally manipulated observed concentrations to estimate the spatial and temporal impact from the Maumee River plume. Using this delineated Maumee River impact in time and space we tested the hypothesis that when water movement is incorporated, there is a linear relationship between river load and western Lake Erie water TP concentrations."

R2C6/C7 Line 254: TP is conservative in the context of this model only if we neglect settling, right? Line 256: I'm assuming that the dilution effect that is mentioned here is due to the increasing depth? What would be magnitude of dilution one would expect, given the known bathymetry? So, given that settling and dilution are known processes that would reduce TP concentration, why does the model still yield beta values that are 1?

Our model of TP is conservative to settling and dilution specifically in our extant within Western Lake Erie. While both settling and dilution are happening within the lake our model lacks the requisite detail to represent it and these effects are therefore subsumed by the data and process error leaving the coefficients close to 1. A more mechanistic process model which defined the effect of dilution and settling is an appropriate next step to our model and we hope that this manuscript prompts such models. We will adjust the text to make this more apparent

Lines 276 - 286

TP is a conservative water quality constituent. TP observations are insensitive to biogeochemical transformations of phosphorus form because these data represent both the organic and inorganic

forms of phosphorus occurring in the water column. $\beta_{mau}$, $\beta_{rai}$, $\beta_{self}$, and $\beta_{lake}$ near 1 would then be expected in the absence of dilution and settling. $\beta$s larger than 1 would indicate in-lake sources of TP. Every $\beta_{mau}$, $\beta_{ras}$, $\beta_{lake}$, and $\beta_{self}$ fit in our models had 95% predictive intervals encompassing a value of 1. However, dilution, settling, and internal loading of TP are happening within our modeled extent in western Lake Erie. Our model lacks the specificity to capture dilution, settling, and internal loading and therefore their effect is being accumulated in the error terms. However, this state space framework could be defined with a mechanistic process model that did capture these effects. Additionally, while our framework could be implemented with the coefficients ($\beta$Mau, $\beta$Ras, $\beta$Lake, and $\beta$Self) fit hierarchically by year potentially defining the overall effect of dilution, settling, and internal loading, current restrictions on computer memory prevented that use here. However, for smaller spatial and temporal models it could be effective.

---

## Author Response (AR3)

Thank you for your reply to reviewer comments and manuscript revisions. I believe that the manuscript has been substantially improved and clarified during the review process and I hope that you found the reviewer comments helpful. I have a few remaining suggestions.

Thank you for the opportunity to revise our manuscript. Below are the changes based on your comments, our comments are in green, text added to the manuscript are blue, and your original comments are in black. Please note that line numbers indicated on this document reflect those of the "track changes" version of the revised manuscript. The authors also made small grammatical corrections and added the Great Lakes Restoration Initiative to the acknowledgement section, noted in the tracked changes document as "Author edits".

E1C1   R1C7 In your reply to reviewer comment, you state that truncating the values of the latent state does not set a prior based on the data, rather it constrains the later state to plausible values. Does this point relate to your explanation of the high Bayesian R2 values (in response to R1C4), whereby 'the model passes through the observed data and the daily observation in the river datasets are likely driving Bayesian R2 values higher with their constrained predictive intervals'? You also say that the Bayesian R2 may not be a fair metric to evaluate the model performance. Hence, is there a link between the Reviewer's point that such constraints (or priors) should not be informed by observations to which the data is fitted and your explanation of the high Bayesian R2 values? Or perhaps I have misunderstood. However, if there is a link, could this be reflected in the discussion as a potential limitation and area for future improvement?

Our reasoning that the upper truncation on the latent values is not associated with the anomalously high Bayesian R2 values. The upper truncation is based on the condition that no concentration observed in the lake will ever be greater than the largest concentration in the Maumee River.

Lines 148 to 149

The maximum observed Maumee River concentration was used as the upper truncating value because no observation in the lake will exceed this value.

Figure 2(a-f) shows why our Bayesian R2 values are likely not an accurate representation of model fit. River samples (a) and (d) contain the bulk of our data, thus the model passing through those daily values does a great job fitting posteriors with constrained predictive intervals. While (b,c,e,and f) show how difficult it is to fit values out in the lake where data is sparse. The posterior predictive p-values and leave-one-out metrics of model fit specifically address these places where data is sparse making up for the Bayesian R2 values which are not telling a complete story. We have added a reference to Figure 2a & 2d to the text we inserted during the previous revision at Line 277.

[Figure]

Cropped Figure 2(a-f).

**E1C2  R1C8** I found the format suggested by the reviewer to explain the model clear and helpful. Can you please take another look at the wording proposed by the reviewer and consider adopting it, or parts of it, in the manuscript? I believe it makes the model structure clear and explicit, especially the explanation of the two models – data=observation and latent=process.

We have amended and added text to the manuscript based on the previous responses.

Lines 124 - 167

The distance between each daily offset surface current location ($Lat_1$, $Lon_1$) and each 2 km-by-2 km concentration node was measured and the node $n$ with the shortest distance defined the adjacency matrix to associate each node $n$ on day $t$ with the node $k$ on day $t$-$1$. The state-space model consists of two models, an observation model of data (y) and a latent state (x) process model.

$$y_{n,t,y} \sim N\left(x_{n,t,y}, \sigma^2\right) \tag{5}$$

Log-transformed TP concentration observations ($y$) at the $n^{th}$ node on the $t^{th}$ day of the $y^{th}$ year was estimated with a normal data model sampled from the unobserved latent state variable ($x$) at the $n^{th}$ node on the $t^{th}$ day of the $y^{th}$ year with standard deviation $\sigma$ (Eq 5). The process model is a first order Markov, only depending on the value of the node at time $t$-$1$ which transported TP to node n at time $t$, that source node is denoted $k$. For nodes in the river, $k=n$ and for nodes in the lake, $k$, is determined from the time $t$ adjacency matrix.

$$x_{n,t,y} \sim \text{Truncated } N\left(f\left(x_{n,t-1,y}\right), \tau^2\right) \quad I(a \le x_{n,t,y} \le b) \tag{6}$$

where

$$f\left(x_{n,t-1,y}\right) = \begin{cases} x_{k,t-1,y} * \beta_{mau} & if\ n = Maumee\ River\ Node \\ x_{k,t-1,y} * \beta_{rai} & if\ n = River\ Raisin\ Node \\ x_{k,t-1,y} * \beta_{self} & if\ n = same\ lake\ Node \\ x_{k,t-1,y} * \beta_{lake} & if\ n = different\ lake\ Node \end{cases} \tag{7}$$

The latent state ($x_{n,t,y}$; Eq 6) is sampled from a normal distribution of a predicted latent state ($f(x_{n,t-1,y})$, Eq 7) and standard deviation $\tau$. $x_{n,t,y}$ was truncated by the detection limit of TP laboratory analysis (5 µg l$^{-1}$, $a$, Eq 6) and the maximum value observed in each year ($y$) within the Maumee River ($b$, Eq 6). The maximum observed Maumee River concentration was used as the upper truncating value because no observation in the lake will exceed this value. Priors were uninformative and defined as;

$$\sigma, \tau \overset{iid}{\sim} Gamma(0.001, 0.001)$$

$$\beta_{self}, \beta_{lake} \overset{iid}{\sim} Normal(0, 10{,}000)$$

$$\beta` \sim Normal(0, 10{,}000)$$

$$\tau`_{mau}, \tau`_{rai} \overset{iid}{\sim} Gamma(0.001, 0.001)$$

$$\beta_{mau} \sim Normal(\beta`, \tau`^2_{mau})$$

$$\beta_{rai} \sim Normal(\beta`, \tau`^2_{rai})$$

River model coefficients ($\beta_{mau}$ and $\beta_{rai}$ ) were fit hierarchically ($N(\beta`, \tau`^2)$) because the ecological and anthropogenic processes enacted on these watersheds are similar, if at different scales. The two lake models were fit with two independent $\beta$ coefficients depending on if the nearest adjacent node $k$ is the same as the estimated node $n$ ($\beta_{self}$) or if a different node $k$ is the nearest ($\beta_{lake}$). Separate independent in-lake models were used to capture different potential drivers of TP concentration through time depending on whether each node was subject to little surface water movement ($\beta_{self}$) or active surface water movement ($\beta_{lake}$).

E1C3   R1C9 I think this explanation is of interest and would be helpful to include in the manuscript for a better understanding of the physical processes operating on the lake.

We have amended and added text to the manuscript based on the previous responses.

Lines 303 - 304

The lack of $\sigma$ and $\tau$ correlation was also visually assessed (Figure C1(c)).

Lines 356 - 361

Distance and direction across our model extent is wrapped up in the change observed in predictive intervals through time. In our framework distance and whether the water mass from the Maumee River physically moves toward a node combine. There are several days in which even the nodes closest to the Maumee River are bypassed because the currents take Maumee River water in a different direction. The dual complications of distance and movement have complicated previous attempts at defining a single relationship between Maumee River load and observed in-lake concentrations, which we overcome here.

E1C4  R2C1and R2C2 I believe that your response to these reviewer questions may be of interest to the readers. Can you please consider including these points in the manuscript?

We have amended and added text to the manuscript based on the previous responses.

Lines 283 – 300

TP is a conservative water quality constituent. TP observations are insensitive to biogeochemical transformations of phosphorus form because these data represent both the organic and inorganic forms of phosphorus occurring in the water column. $\beta$mau, $\beta$ras, $\beta$lake, and $\beta$self fit in our models had 95% predictive intervals encompassing a value of 1. Coefficients were close to 1 because on our daily time-step, the TP concentrations do not widely vary (e.g., the concentration today is similar to the concentration yesterday). The uncertainty in the process and data models allows the model predictions to trend toward the observations where available and be constrained where previous time-steps passed through observations. Were these coefficients to exceed 1 this would be evidence of other inputs of P or less that 1 would indicate some internal loss such as settling or dilution. TP is conservative to processes within the water column because it accounts for the dissolved and particulate P, if our model was applied only to dissolved P which is subject to strong assimilation pressure by phytoplankton the model coefficients would likely be negative. While dilution, settling, and internal loading of TP are happening within our modelled extent in western Lake Erie. Our model lacks the specificity to capture dilution, settling, and internal loading and therefore their effect is being accumulated in our error terms. However, this state-space framework could be defined with a mechanistic process model that did capture these effects. Additionally, while our framework could be implemented with the coefficients ($\beta$Mau, $\beta$Ras, $\beta$Lake, and $\beta$Self) fit hierarchically by year potentially defining the overall effect of dilution, settling, and internal loading, current restrictions on computer memory prevented that use here. However, for smaller spatial and temporal models it could be effective.

Lines 322 – 333

Our model framework allows information from discrete grab samples to be shared across any waterbody where the movement pattern of water is available. Additionally, this model can generate estimates at unobserved nodes or at unobserved time-steps of observed nodes without requiring defined biogeochemical processes of a mechanistic model. For our application, the 2km x 2km grid was chosen to match the surface current dataset. While we did not experiment with other discrete grid distances, any applicable configuration will work. Defining a reasonable grid distance could be based on the spatial distribution of the available data and the user's willingness to extrapolate or average surface current direction and magnitude. Similarly, our temporal time-step was daily, but this could also be applied to monthly data in data sparse systems or hourly data in data rich applications. This spatial and temporal flexibility or using state space frameworks gives users the capacity to tune the computational runtime and resolution of models to fit the hypothesis tested.